



# Output-constrained individual pitch control methods using the multiblade coordinate transformation: Trading off actuation effort and blade fatigue load reduction for wind turbines

Jesse I.S. Hummel[1], Jens Kober[1], and Sebastiaan P. Mulders[1]

[1]Delft University of Technology, Mekelweg 5, 2628 CD Delft, the Netherlands

**Correspondence:** Jesse I.S. Hummel (j.i.s.hummel@tudelft.nl)

**Abstract.** Individual pitch control (IPC) has been thoroughly researched for its ability to reduce wind turbine blade and tower fatigue loads. Conventional IPC often uses the multiblade coordinate (MBC) transformation and aims for full attenuation of the oscillating loads. However, this also leads to high control effort and increased fatigue damage on the pitch system. Output-constrained IPC uses the minimum actuator effort to drive loads to some reference value instead of fully attenuating them, achieving a trade-off between load reduction and actuator effort. To date, no control method exists that achieves output-constrained IPC using the conventional MBC approach. Furthermore, while multiple constrained IPC approaches have been proposed and analyzed, none of them analyze the full range of operating points between 'no IPC' and 'full IPC'. This paper presents two output-constrained IPC methods that use the MBC transformation. The first method, $\ell^\infty$-IPC, independently drives the tilt and yaw moment to a tilt and yaw reference, while the second method, $\ell^2$-IPC, directly targets the magnitude of the combined tilt and yaw load. We furthermore analyze all operating points between no IPC and full IPC. OpenFAST simulations of the IEA 15 MW turbine were run at a wind speed of 15 m/s. In laminar conditions, $\ell^2$-IPC is more efficient because it reduces the magnitude of the load directly, while $\ell^\infty$-IPC also uses control effort to change the phase of the blade load in the direction of the load references. To assess the performance in realistic wind conditions, results are averaged over multiple turbulent wind seeds. Both $\ell^\infty$-IPC and $\ell^2$-IPC have a similar performance and the operating points between no IPC and full IPC form a nonlinear trade-off. One of the operating points in this trade-off achieves a 50% load reduction, measured in damage equivalent load, with just 16.4% of the actuator effort, measured in actuator duty cycle, compared to conventional IPC with the same integrator gain. This shows the potential of output-constrained IPC to facilitate a superior trade-off between load reduction and actuator effort.

## 1 Introduction

Wind energy has become a mainstream energy provider due to its severe cost reduction in recent decades. This cost reduction was, in part, driven by engineering efforts that enabled taller towers, longer blades, and higher capacity factors of wind turbines





(Veers et al., 2019). However, longer blades are more flexible and sample a larger area of the spatially and temporally varying wind field, making them susceptible to fatigue loading and causing new challenges.

Individual pitch control (IPC) can alleviate some of these fatigue loads by pitching the three blades independently to reduce the periodic loading that arises from wind shear, tower shadow, turbulence, and rotor misalignment (Bossanyi, 2003). This could decrease the material cost of the wind turbine and/or increase their lifespan (Pettas et al., 2018).

IPC implementations using classical control techniques often use the multiblade coordinate (MBC) transformation (Bir, 2008), also called the Park or d-q transformation originating from electrical engineering (Park, 1929), or Coleman transforma-
tion originating from helicopter rotor control (Coleman and Feingold, 1958). The MBC transformation converts signals from a rotating reference frame to a stationary reference frame, and vice versa. In the context of three-bladed wind turbines, it transforms a set of three signals from the rotating blade frame to a set of three signals in the nonrotating frame, usually referred to as collective, tilt, and yaw components, that represent the rotor as a whole. By converting the blade dynamics to rotor dynamics in the nonrotating frame, they can be analyzed together in a common reference frame with the other nonrotating components,
such as the tower (Bir, 2008).

The MBC transformation can also be used to control oscillations in blade loads. The forward transformation converts these loads to steady tilt and yaw moments in the nonrotating frame. These tilt and yaw moments are then driven to zero by two independent SISO controllers, typically integral (I) or proportional-integral (PI) controllers, that produce a tilt and yaw pitch signal. These outputs are subsequently transformed back to the rotating frame using the inverse MBC transformation, resulting
in a sinusoidal pitch signal that has a 120° offset between each blade (Bossanyi, 2003, 2005; van Solingen and van Wingerden, 2015). The dominant load on the blades is 1P (once-per-revolution), mainly due to wind shear. This 1P oscillation can be reduced by a 1P IPC controller. However, the 2P harmonic can also be reduced by using a 2P MBC transformation. While the 2P harmonic typically contributes less to the fatigue damage of the blade, since their magnitude is smaller, they contribute to fatigue damage on the tower since the 2P blade load maps to a 3P oscillation on the tower (van Solingen and van Wingerden,
2015; Bossanyi et al., 2013).

Due to the rotation of the blades and system dynamics arising from, e.g., actuator and blade dynamics, the tilt and yaw axes are coupled (Mulders et al., 2019). For flexible wind turbines, these dynamics are slower and thus lead to a strong coupling between the tilt and yaw axes. This coupling necessitates a multivariable controller design (Bossanyi, 2003; Lu et al., 2015) or decoupling of the tilt and yaw axes, enabling SISO controller design. This can be achieved by using an azimuth offset in
the inverse MBC transformation, as was already noted by Bossanyi (2003) and formally analyzed by Mulders et al. (2019); Mulders and van Wingerden (2019), who also provide a method to find the optimal azimuth offset.

Several field tests have been carried out, demonstrating the load reduction capability of IPC (Bossanyi et al., 2013; Shan et al., 2013; Van Solingen et al., 2016; Ossmann et al., 2021). However, since conventional IPC always aims for complete load alleviation, it leads to excessive pitch actuation. This has hindered industry adoption (Novaes Menezes et al., 2018). A control
method that balances load reduction and pitch actuation would make IPC more practically feasible.





Three methods to constrain IPC controllers can be distinguished that enable the trade-off between load reductions and pitch activity: input-constrained IPC, where the actuator angle, rate, and/or acceleration is limited; output-constrained IPC, where the amplitude of the oscillating loads is regulated; and fatigue-constrained, which aims to directly constrain the fatigue damage.

Input-constrained IPC can be implemented when using the MBC transformation by realizing that the resulting blade pitch signal is sinusoidal, of which the maximum, maximum rate, and maximum acceleration can be derived using the amplitude of the pitch signal in the nonrotating domain and rotor speed. Bossanyi (2005) introduces a limit schedule that limits the output of the PI controller in the nonrotating frame, thus limiting the pitch angle for each blade. Kanev and van Engelen (2009) extend this and use an anti-windup strategy to limit the pitch angle, rate, and acceleration. The previous studies assume a constant rotor speed while Ungurán et al. (2019) also derive angle and rate limits for non-constant rotor speeds. They furthermore note that adding a rate limit causes a time lag in the pitch signal, which slightly reduces the efficiency of IPC.

Input constraints can also be implemented when using a model predictive control (MPC) method by including additional constraints in the optimization problem that is solved during each timestep. Raach et al. (2014) include actuator constraints in a nonlinear MPC framework. Petrović et al. (2021) propose a method to convexify these constraints to get a convex optimal control problem when using MPC. Lastly, Liu et al. (2021) include input constraints in a constrained subspace predictive repetitive control (cSPRC) framework, a data-driven MPC controller.

Output-constrained IPC sets limits on the permissible loads and employs the minimal pitch signal required to maintain the loads within these limits, thus opting for no IPC action when the loads naturally fall within these limits. Liu et al. (2022) later used the same cSPRC framework to constrain the periodic blade loads rather than the pitch angles to form an output-constrained IPC method. Henry et al. (2024) uses the MBC transformation and sets a positive reference on the tilt axis, thus only constraining the positive tilt load and assuming a negligible yaw moment.

Fatigue-constrained IPC sets direct bounds on the fatigue damage that may be accumulated. Since the fatigue calculation is usually algebraic and highly nonlinear, Collet et al. (2020) derive a convex, data-driven objective function that can approximate the fatigue damage on both the blades and the actuators. Their MPC controller can find a trade-off between pitch actuation and actuator activity by weighting the fatigue damage differently for the blade and actuator fatigue damage.

Comparing these methods, we see that output-constrained IPC sets limits on permissible loads and employs the minimal pitch signal required to maintain these loads within the specified bounds. When the loads naturally fall within these reference bounds, the controller refrains from IPC action. In contrast, input-constrained IPC consistently actuates the pitch but never exceeds a maximum pitching angle, rate, or acceleration. Fatigue-constrained IPC poses challenges due to the nature of fatigue damage calculation. The conventional approach of rainflow counting to get a damage equivalent load cannot be done in real time, thus requiring estimation techniques.

Consequently, output-constrained IPC emerges as a promising method to balance load reduction and pitch actuation. Furthermore, the load references could be integrated into the wind turbine design process, thus enabling control co-design of the wind turbine (Pao et al., 2024) with IPC control, realizing its potential for material reduction. However, little research has focused on this approach, and some output-constrained IPC methods rely on data-driven techniques that are not commonly used in industry or only constrain a positive tilt moment. Additionally, while any constrained IPC control method can explore the





operation space between no IPC and full IPC, the complete trade-off, to the best of our knowledge, has not been investigated. Notably, though, both Han and Leithead (2015) and Lara et al. (2024) have explored a small portion of this space by adjusting the gains of IPC controllers, thus showing the trade-off when operating close to full IPC.

This work explores the full operating region between no and full IPC using two different output-constrained IPC controllers using classical control elements and the MBC transformation. The first method, $\ell^{\infty}$-IPC, constrains the tilt and yaw moments independently, while the second method, $\ell^2$-IPC, constrains the magnitude of the combined tilt and yaw moment. Thereby providing the following contributions:

1. Proposing the two output-constrained control methods $\ell^{\infty}$-IPC and $\ell^2$-IPC.
2. Deriving the original load estimator, used by both control methods.
3. Sharing the two output-constrained control methods in an open and freely-accessible online repository (Hummel, 2024).
4. Analyzing the working mechanism of these controllers in laminar conditions.
5. Analyzing the trade-off between fatigue load and pitch actuation when operating at any operating point between no and full IPC both in laminar and turbulent flow conditions.

In this work, we refer to the IPC methods that aim for full load alleviation as conventional, unconstrained, or full IPC. When only the collective pitch controller is active, this is referred to as no IPC. Furthermore, this work focuses on blade loads only, and thus only on 1P IPC. However, by adding additional MBC loops, the proposed control methods could be easily extended to higher harmonics.

This paper proceeds as follows: First, the MBC transformation and its use for IPC is discussed in Sect. 2. An illustrative example of using controller tuning with conventional IPC to achieve a trade-off between load reduction and actuator effort is given in Sect. 3. Next, the two output-constrained IPC controllers are introduced in Sect. 4. The results of the controllers are presented in Sect. 5. Finally, conclusions are drawn in Sect. 6.

## 2 Individual pitch control using the multiblade coordinate transformation

This section provides an overview of individual pitch control using the MBC transformation. Besides general theory, the rotations that lie at the fundament of the MBC transformations are shown, and how these rotations are used to decouple the orthogonal tilt and yaw axes using an azimuth offset is discussed.

### 2.1 Motivation behind the MBC transformation

The MBC transformation transforms a set of three signals from a rotating reference frame to a nonrotating reference frame, or vice versa. It has proven helpful in two ways in the wind turbine field: system analysis and control. First, by transforming the dynamics from the rotating blades to a nonrotating frame, the dynamics of the rotor are unified with the dynamics of the rest of the wind turbine, enabling physical insights of this interaction (Bir, 2008). Second, it can be used for individual pitch control. By transforming the oscillating blade loads to a nonrotating frame, the targeted harmonic is converted to a steady-state signal, which greatly simplifies controller design and implementation (Bossanyi, 2003).



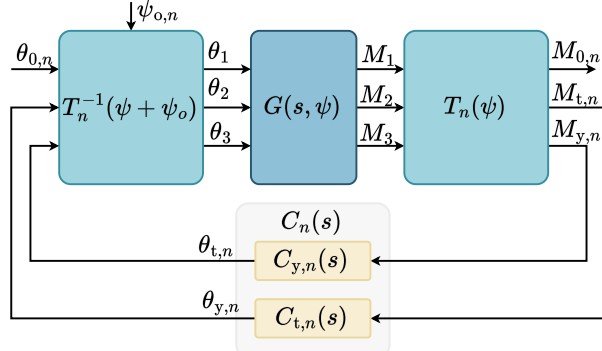

**Figure 1.** Block diagram of conventional IPC for the $n$P blade harmonic, fully attenuating the tilt and yaw moments. The forward MBC transformation $\mathbf{T}_n$ transforms the rotating blade loads to the nonrotating frame, where the SISO controllers $C_{\mathrm{t},n}$ and $C_{\mathrm{y},n}$ fully attenuate the tilt and yaw moments. The inverse MBC transformation $\mathbf{T}_n^{-1}$ converts the pitch signals back to the rotating frame and includes an azimuth offset $\psi_{\mathrm{o},n}$ to decouple the tilt and yaw axes.

The periodicity of the blade load is caused by wind shear, tower shadow, and rotor misalignment (Bossanyi, 2003). Due to the periodic nature of the system, the blade load power spectrum mainly consists of $n$P harmonics, where $n \in \mathbb{N}$, of the

125 rotation of the rotor (defined as 1P, or once-per-revolution). Only stochastic disturbances, such as turbulence, cause power at non-integer multiples of the rotation frequency. When using the MBC transformation, the $n$P harmonics map to the nearest $3n$P harmonic in the nonrotating frame, while the $3n$P harmonic itself is filtered out. So, the 1P blade load is experienced as a 0P (constant) load in the nonrotating frame, the 2P and 4P blade load as a 3P nonrotating load, and so on (van Solingen and van Wingerden, 2015).

Since the 1P blade loads are dominant (Bossanyi, 2005), most research focuses on damping these loads. However, since the 2P blade loads cause a 3P load in the nonrotating frame, dampening the 2P blade loads also reduces fatigue loads on the tower (van Solingen and van Wingerden, 2015). This work focuses on the 1P blade loads. However, the proposed control methods can be easily extended to higher harmonics by adding additional MBC loops at those harmonics (Mulders and van Wingerden, 2019).

**2.2 General MBC-IPC theory**

An IPC implementation for blade harmonic $n$ using the MBC transformation is shown in Figure 1. The plant $\mathbf{G}(s,\psi)$, where $\psi \in [0, 2\pi)$ is the azimuth angle of the rotor, is a linear parameter-varying (LPV) plant due to azimuth dependent effects on the wind turbine dynamics, such as gravity. The three blade loads $\boldsymbol{M}_b$, where $b \in \{1, 2, \ldots, B\}$ and $B = 3$, are transformed to a nonrotating coordinate system using the forward MBC transformation defined as

$$\boldsymbol{M}_{\mathrm{N},n}(t) = \mathbf{T}_n(\psi(t))\boldsymbol{M}_{\mathrm{R}}(t), \tag{1}$$



with

$$
\mathbf{T}_n = \frac{2}{B} \begin{bmatrix} 1/2 & 1/2 & 1/2 \\ \cos(n\psi_1(t)) & \cos(n\psi_2(t)) & \cos(n\psi_3(t)) \\ \sin(n\psi_1(t)) & \sin(n\psi_2(t)) & \sin(n\psi_3(t)) \end{bmatrix},
$$

where $\boldsymbol{M}_{\mathrm{N},n} = \begin{bmatrix} M_{0,n} & M_{\mathrm{t},n} & M_{\mathrm{y},n} \end{bmatrix}^{\mathsf{T}}$ denotes the nonrotating moment vector for the $n$P harmonic consisting of the collective, tilt, and yaw flapping moments, $\psi_b$ the azimuth position of blade $b$, and $\boldsymbol{M}_{\mathrm{R}} = \begin{bmatrix} M_1 & M_2 & M_3 \end{bmatrix}^{\mathsf{T}}$ the rotating moment vector consisting of the flapping moments of the three blades in the rotating frame.

If the blade load $\boldsymbol{M}_{\mathrm{R}}$ contains three pure $n$P harmonic sine waves, $\boldsymbol{M}_{\mathrm{N},n}$ is a steady-state signal. However, due to the presence of other harmonics in the blade loads, $\boldsymbol{M}_{\mathrm{N},n}$ also contains power at $3n$P harmonics. Furthermore, for unbalanced rotors, arising from load imbalance or pitch imbalance, the signal contains power at all $n$P harmonics (van Solingen and van Wingerden, 2015). In addition, turbulence and other stochastic effects give $\boldsymbol{M}_{\mathrm{N},n}$ power at any frequency. To focus the control loop on a single harmonic, $\boldsymbol{M}_{\mathrm{N},n}$ is usually low-pass filtered, together with a 3P notch filter (Bossanyi, 2005; van Solingen and van Wingerden, 2015), so that only the 0P contribution in the nonrotating frame, and thus the $n$P contribution in the rotating frame, is attenuated.

Usually, two I or PI-controllers are implemented in a diagonal controller configuration, represented as

$$
\boldsymbol{\theta}_{\mathrm{N},n}(s) = \begin{bmatrix} 0 & 0 & 0 \\ 0 & C_{\mathrm{t},n}(s) & 0 \\ 0 & 0 & C_{\mathrm{y},n}(s) \end{bmatrix} \boldsymbol{M}_{\mathrm{N},n}(s) \tag{2}
$$

where $\boldsymbol{\theta}_{\mathrm{N},n} = \begin{bmatrix} \theta_{0,n} & \theta_{\mathrm{t},n} & \theta_{\mathrm{y},n} \end{bmatrix}^{\mathsf{T}}$ denotes the nonrotating pitch vector consisting of the collective, tilt, and yaw pitch angles of the $n$P harmonic in the nonrotating frame and denotes the pitch angles in the nonrotating frame. The IPC controller is only active on the tilt and yaw channels, which relate to the oscillating part of the moments, so the first row and column are filled with zeros. These controllers fully attenuate the tilt and yaw moments by producing the necessary tilt and yaw pitch angles to drive the tilt and yaw moments to zero. In this work, we refer to this as unconstrained, conventional, or full IPC.

These tilt and yaw pitch angles are then converted back to the rotating frame using the inverse MBC transformation, defined as

$$
\boldsymbol{\theta}_{\mathrm{R}}(t) = \mathbf{T}_n^{-1}(\psi(t) + \psi_{\mathrm{o},n}) \boldsymbol{\theta}_{\mathrm{N},n}(t), \tag{3}
$$

with

$$
\mathbf{T}_n^{-1} = \begin{bmatrix} 1 & \cos(n[\psi_1(t) + \psi_{\mathrm{o},n}]) & \cos(n[\psi_1(t) + \psi_{\mathrm{o},n}]) \\ 1 & \cos(n[\psi_2(t) + \psi_{\mathrm{o},n}]) & \cos(n[\psi_2(t) + \psi_{\mathrm{o},n}]) \\ 1 & \cos(n[\psi_3(t) + \psi_{\mathrm{o},n}]) & \cos(n[\psi_3(t) + \psi_{\mathrm{o},n}]) \end{bmatrix},
$$

where $\boldsymbol{\theta}_{\mathrm{R}} = \begin{bmatrix} \theta_1 & \theta_2 & \theta_3 \end{bmatrix}^{\mathsf{T}}$ denotes the rotating pitch vector containing the pitch angles of the three blades defined in the rotating frame and $\psi_{\mathrm{o},n} \in \mathbb{R}$ denotes the azimuth offset used for the $n$P harmonic.





### 2.3 Rotations in the MBC transformation

The MBC transformation can be decomposed as a Clarke transformation followed by a rotation (O'Rourke et al., 2019). Note that the authors define this rotation as the DQ0 transformation, but the DQ0 transformation is usually defined as equal to the MBC transformation.

An offset in the forward or inverse MBC transformation results in an offset in the rotation transformation and thus a rotation of the nonrotating frame. This is mathematically derived by considering the forward MBC transformation for the first harmonic with an azimuth offset $\psi_\mathrm{r}$, given by

$$
\begin{aligned}
&\mathbf{T}(\psi + \psi_\mathrm{r}) \\
&= \frac{2}{3}
\begin{bmatrix}
1/2 & 1/2 & 1/2 \\
\cos(\psi_1 + \psi_\mathrm{r}) & \cos(\psi_2 + \psi_\mathrm{r}) & \cos(\psi_3 + \psi_\mathrm{r}) \\
\sin(\psi_1 + \psi_\mathrm{r}) & \sin(\psi_2 + \psi_\mathrm{r}) & \sin(\psi_3 + \psi_\mathrm{r})
\end{bmatrix} \\
&= \frac{2}{3}
\begin{bmatrix}
1 & 0 & 0 \\
0 & \cos(\psi_\mathrm{r}) & -\sin(\psi_\mathrm{r}) \\
0 & \sin(\psi_\mathrm{r}) & \cos(\psi_\mathrm{r})
\end{bmatrix} \\
&\quad \cdot
\begin{bmatrix}
1/2 & 1/2 & 1/2 \\
\cos(\psi_1) & \cos(\psi_2) & \cos(\psi_3) \\
\sin(\psi_1) & \sin(\psi_2) & \sin(\psi_3)
\end{bmatrix} \\
&= R(\psi_\mathrm{r})\mathbf{T}(\psi),
\end{aligned}
\tag{4}
$$

where $R(\psi_\mathrm{r})$ is a rotation matrix rotating the nonrotating frame by $\psi_\mathrm{r}$ around the collective axis. A similar derivation holds for the inverse MBC transformation.

### 2.4 Frequency domain analysis

The MBC transformations can be transformed to the Laplace domain, enabling frequency domain analysis, useful for controller design and calibration. The main results from Lu et al. (2015) and Mulders et al. (2019) are given here. Transforming the forward transformation (Eq. (1)) to the Laplace domain yields

$$
\boldsymbol{\mathcal{M}}_{\mathrm{N},n}(s) = \frac{2}{3}C_{\mathrm{L},n}\boldsymbol{\mathcal{M}}_{\mathrm{R}}(s_-) + \frac{2}{3}C_{\mathrm{H},n}\boldsymbol{\mathcal{M}}_{\mathrm{R}}(s_+),
\tag{5}
$$

and the inverse transformation (Eq. (3)) is transformed to

$$
\boldsymbol{\theta}_{\mathrm{R}}(s) = \tilde{C}^{\mathsf{T}}_{\mathrm{L},n}(\psi_{\mathrm{o},n})\boldsymbol{\theta}_{\mathrm{N},n}(s_-) + \tilde{C}^{\mathsf{T}}_{\mathrm{H},n}(\psi_{\mathrm{o},n})\boldsymbol{\theta}_{\mathrm{N},n}(s_+),
\tag{6}
$$

where $C_{\mathrm{L},n}$ and $C_{\mathrm{H},n}$ denote the *low* and *high* partial transformation matrices for the $n\mathrm{P}$ harmonic respectively, $\tilde{C}^{\mathsf{T}}_{\mathrm{L},n}(\psi_{\mathrm{o},n})$ and $\tilde{C}^{\mathsf{T}}_{\mathrm{H},n}(\psi_{\mathrm{o},n})$ the transpose of the partial transformation matrices including azimuth offset for the $n\mathrm{P}$ harmonic, and $s_\pm = s \pm \mathrm{j}n\omega_\mathrm{r}$ and denotes the frequency-shifted Laplace operator.





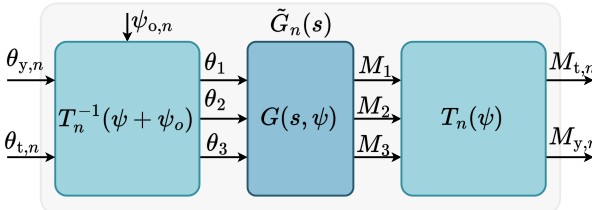

**Figure 2.** Block diagram of the demodulated plant $\tilde{G}_n(s)$. By calibrating the azimuth offset $\psi_{\mathrm{o},n}$, the low-frequency cross-coupling from $\theta_{\mathrm{t}}$ and $\theta_{\mathrm{y}}$ to $M_{\mathrm{y}}$ and $M_{\mathrm{t}}$ respectively, is minimized.

By analyzing the wind turbine plant $G(s,\psi)$ surrounded by the MBC transformations, the demodulated plant $\tilde{G}_n(s)$ for the $n$P harmonic is defined, shown in Figure 2. This demodulated plant is linear time-invariant (LTI) after averaging out the azimuth dependency, which physically makes sense since the rotating blade loads are transformed to a nonrotating frame, where the azimuth angle has no meaning (Bir, 2008). Note that the demodulated plant is depicted as a $2 \times 2$ system without the collective input $\theta_0$ and collective output $M_0$ since these are not linked to the periodic loads and are not used by the individual pitch controller.

Due to dynamics in the wind turbine, there is a coupling between the tilt and yaw inputs and outputs. The system thus needs to be decoupled to enable the use of SISO controllers.

## 2.5 Decoupling using the Optimal Azimuth Offset

Phase lag in the system, due to, e.g., actuator dynamics, blade dynamics, dynamic induction, and communication delays, causes coupling in the nonrotating frame, so from the tilt input to the yaw output, and vice versa. Especially flexible wind turbines have slower dynamics, and thus more coupling, which must be considered during the controller design process.

The literature describes two ways to deal with this coupling. The first is to design a multivariable controller that takes the coupling into account (Bossanyi, 2003; Lu et al., 2015). Second, the system can be decoupled by using an azimuth offset in the inverse MBC transformation (Mulders et al., 2019; Mulders and van Wingerden, 2019). After the system has been decoupled, SISO controllers are used to independently control the tilt and yaw moments.

The demodulated plant $\tilde{G}_n(s)$ is fully decoupled when its relative gain array (RGA) (Skogestad and Postlethwaite, 2001) is close to identity. The RGA is a measure of the coupling between the inputs and outputs of a system and is defined as

$$\mathbf{R}(\mathrm{j}\omega) = \mathbf{H}(\mathrm{j}\omega) \circ \mathbf{H}(\mathrm{j}\omega)^{-\mathsf{T}}, \tag{7}$$

where $\mathbf{H}$ denotes the frequency response matrix of the system, $\omega$ the frequency, and $\circ$ element-wise multiplication.

The azimuth offset allows the system to be decoupled in the low-frequency region and works by introducing an offset between the forward and inverse rotations, discussed in Sect. 2.3. Decoupling at low frequencies is sufficient and effective because the IPC controller is only active in this region. The optimal azimuth offset is found by minimizing the RGA of the off-diagonal components for low frequencies. In this work, the level of low-frequency coupling is defined as the highest off-



diagonal element of the RGA matrix averaged over the low frequencies and is given by

$$R_\# = \max_{m \neq n} \left\{ \frac{1}{\omega_\mathrm{m}} \int_0^{\omega_\mathrm{m}} |\mathbf{R}_{m,n}(\mathrm{j}\omega)| \, \mathrm{d}\omega \right\}, \qquad\qquad m, n \in \{\mathrm{t}, \mathrm{y}\}, \quad (8)$$

where $\omega_\mathrm{m}$ denotes the upper limit of the low-frequency range. By selecting $m$ and $n$ in the set $\{\mathrm{t}, \mathrm{y}\}$ and specifying $m \neq n$, the off-diagonal elements from tilt to yaw and yaw to tilt are selected. The optimal azimuth offset is subsequently found by
215 minimizing $R_\#$.

Note that since this work focuses on the 1P blade loads, so $n = 1$. Furthermore, the subscript 1 is omitted for brevity in the rest of this work. The following section uses the MBC transformation with conventional, full IPC to give an illustrative example of the trade-off between load reduction and actuator effort.

## 3 Illustrative example: trading off load reduction and actuator effort with conventional IPC

This section discusses controller objectives, controller calibration, and results from a conventional, unconstrained, full IPC controller. Using different controller gains, the trade-off between actuator effort and load is analyzed, similar to (Han and Leithead, 2015; Lara et al., 2024) but for the IEA 15 MW turbine (Gaertner et al., 2020) at a wind speed of 15 m/s. This will serve as an illustrative example of this trade-off and form the baseline to which the proposed control methods will be compared in Sect. 5.

### 3.1 Control objectives

This work focuses on two conflicting control objectives: fatigue load of the blade in the flapping direction and actuator effort of the pitch actuators. The fatigue load is measured through the damage equivalent load (DEL) (Sutherland, 1999; Thomsen, 1998) and is defined as

$$\mathrm{DEL} = \left( \frac{\sum n_i R_i^m}{n_\mathrm{eq}} \right)^{1/m}, \qquad\qquad (9)$$

where $n_i$ denotes the number of cycles at load level $i$, $R_i$ the load range at load level $i$, $n_\mathrm{eq}$ the number of cycles at the equivalent load level, and $m$ the slope of the S-N curve (or Wöhler slope), typically 10 for composites used for the blade (Zahle et al., 2024). A rainflow counting algorithm (E08 Committee, 2017) is used to obtain $n_i$ and $R_i$ from a flapping moment signal. The DEL is then calculated for a certain number of cycles $n_\mathrm{eq}$, which is typically set to the simulation length to calculate the DEL for a 1 Hz equivalent load. The objective in this work is the flapping moment DEL and is calculated by averaging the flapping
moment DEL for each of the three blades.

The actuator effort is evaluated using the actuator duty cycle (ADC) (Bottasso et al., 2013), which represents a normalized total travel angle and is defined as

$$\mathrm{ADC} = \frac{1}{T} \int_0^T \left| \frac{\dot{u}(t)}{\dot{u}_\mathrm{max}} \right| \mathrm{d}t, \qquad\qquad (10)$$





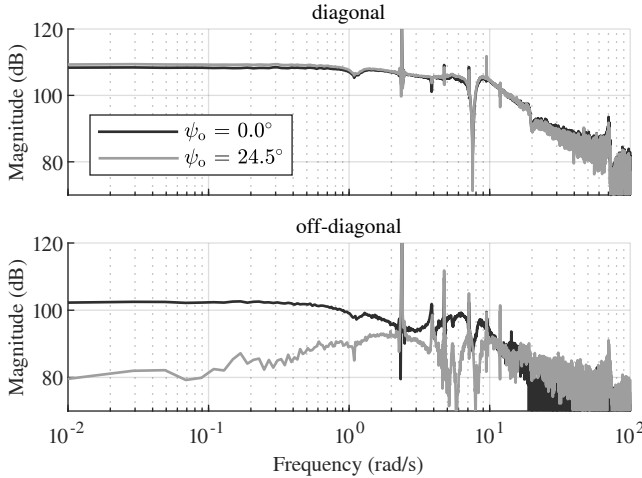

**Figure 3.** Frequency response of the demodulated plant with and without the optimal azimuth offset of 24.5° at a wind speed of 15 m/s. When including the azimuth offset, the off-diagonal gain is significantly reduced at low frequencies, effectively decoupling the system at those frequencies.

where $\dot{u}(t)$ is the time derivative of the control input, $\dot{u}_{\mathrm{max}}$ is the maximum control input rate, and $T$ is the duration over which the ADC is calculated. To calculate the ADC for the wind turbine, the pitch rate is used as the control input, and the average ADC is calculated for the three pitch signals. Furthermore, $\dot{u}_{\mathrm{max}}$ is set to 2°/s, which is the maximum pitch rate of the IEA 15 MW wind turbine (Gaertner et al., 2020).

### 3.2 Controller calibration

This section establishes a properly tuned baseline full IPC controller. Two control variables need to be calibrated, namely the azimuth offset used to decouple the plant and the integrator gain of the IPC controllers.

Using the procedure outlined in Sect. 2.5 of minimizing the highest off-diagonal RGA using the frequency response obtained from a spectral estimate, the optimal azimuth offset for the IEA 15 MW reference turbine at a wind speed of 15 m/s is found to be 24.5°. Figure 3 shows the diagonal and off-diagonal frequency response of the demodulated plant with and without this azimuth offset. The off-diagonal frequency response, in the lower subplot, is significantly reduced at low frequencies when using the optimal azimuth offset, effectively decoupling the system at frequencies where the IPC controllers are active. The response on the diagonal is increased slightly by a few dB. Note that the tilt-to-tilt and the yaw-to-yaw response are almost identical, so the tilt-to-tilt response is shown as the diagonal element. Similarly, the tilt-to-yaw and yaw-to-tilt response are almost identical, and the tilt-to-yaw response is shown as the off-diagonal element.

The diagonal controller, consisting of a tilt and a yaw controller, uses the same gain for both controllers since the diagonals of the decoupled demodulated plant are almost identical. This work assumes a pure I-controller that targets a certain open loop crossover frequency. A grid search is performed to assess the effect of the crossover frequency on the trade-off between DEL

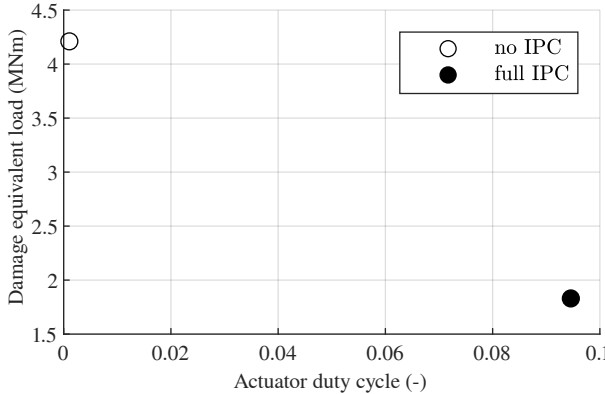

**Figure 4.** Trade-off between DEL and ADC for full IPC in laminar wind conditions. Controller tuning has no effect since the controllers reach a steady-state operating condition. So the only trade-off is to turn IPC on or off.

and ADC. The minimum crossover frequency is set to 0.01 rad/s, which corresponds to a time constant of 100 seconds. At this time constant, the controller would converge to 95% of its steady-state value in 300 seconds, which is the same amount of time that we discard in the simulation to let the turbine reach steady-state. Even slower controllers would take longer to converge, which makes simulating and thus analyzing them impractical. The maximum crossover frequency is set to 1.25 rad/s, at which point the stability margins are significantly reduced, and performance degrades.

## 3.3 Results

Using OpenFAST (Jonkman et al., 2023), simulations were run on the IEA 15 MW reference turbine (Gaertner et al., 2020) in monopile configuration. All degrees of freedom were enabled, but hydrodynamic loads were disabled to focus on the oscillating blade loads induced by wind shear. The simulations were run at a wind speed of 15 m/s so the turbine operates in above-rated conditions with a constant torque and a collective pitch controller implementation from ROSCO (Abbas et al., 2022, 2024) to regulate the rotor speed. A vertical wind shear coefficient of 0.07 with a power law profile was chosen to represent realistic offshore conditions (Yang et al., 2024). Both laminar and turbulent conditions were tested. Each wind condition was run with 9 different integrator gains to assess the effect of controller tuning.

Figure 4 shows the results for laminar wind conditions. All full IPC controllers converge to the same steady-state operating point, and the bandwidth of the controller, as expected, does not affect this. The 1P oscillating load is always completely attenuated since the DC gain of the demodulated plant's diagonal open loop transfer functions is infinite due to the use of an integral control element. This means that the tilt and yaw moments are fully attenuated by significantly pitching the blades and that the controller tuning does not affect the trade-off between DEL and ADC. So there are two options: no IPC with a high fatigue load and low actuator effort or full IPC (with any tuning) with a lower fatigue load and higher actuator effort.

In turbulent wind conditions, the controller constantly adapts to changing wind conditions, so the bandwidth of the controller affects its performance. The turbulent wind fields are generated using TurbSim (Jonkman, 2014) using the IECKAI turbulence

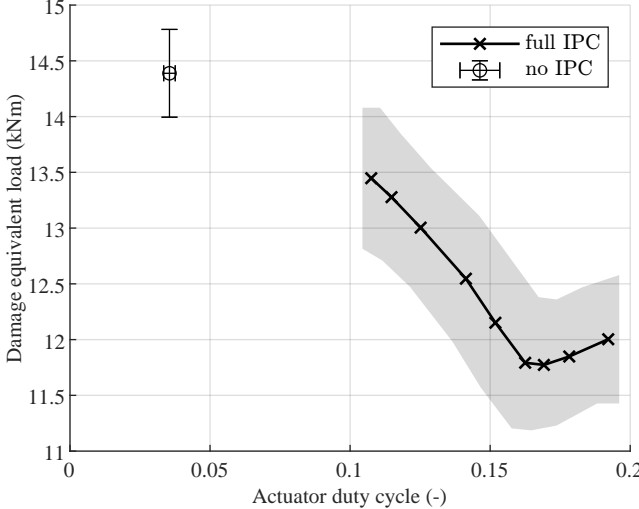

**Figure 5.** Trade-off between DEL and ADC with 8% turbulence intensity averaged over 10 turbulent wind field instances. For a crossover frequency of 0.75 rad/s, the full IPC controller has the highest reduction in DEL, and lowering the gain brings the trade-off closer to no IPC. However, lowering the gain too much results in time constants larger than 100 seconds, so the full IPC controller cannot make the complete trade-off.

model with 8% turbulence intensity. To obtain statistically significant results, each controller tuning was run for 10 different instances of turbulence using a different input seed for TurbSim. Furthermore, each simulation was run for 2100 seconds, resulting in 30 minutes of valid data after discarding the first 300 seconds to eliminate initialization effects.

Over these 30 minutes, the DEL and ADC for each controller gain were calculated and are shown in Figure 5. With no IPC action, there is a high damage equivalent load but little actuator effort. The actuator effort is not equal to zero since the collective pitch control is active and regulates the rotor speed in these turbulent conditions. Furthermore, the fatigue load is higher than in laminar conditions due to the additional excitation of the blades. Conventional, full IPC with different gains results in different trade-offs between DEL and ADC. The highest reduction in load is achieved when targeting a crossover frequency of 0.75 rad/s, and the controllers with higher bandwidth increase the DEL while also increasing the actuator effort. With the lowest bandwidth, the controllers get closer to no IPC, but to get even closer, the time constant would have to be set unreasonably large. Instead of making this trade-off with controller tuning, output-constrained IPC makes this trade-off by setting a reference load. Two such methods are introduced next and compared to this result in Sect. 5.

# 4 Output-constrained individual pitch control using the multiblade coordinate transformation

Conventional, full IPC drives the oscillating loads to zero. Output-constrained IPC instead drives the loads to some reference value. When using the MBC transformation, the tilt and yaw moments are thus driven to some reference. This section defines two types of references through the norm they represent and subsequently proposes two output-constrained IPC methods using





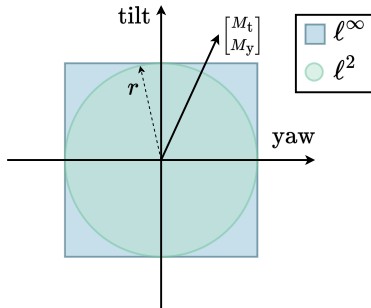

**Figure 6.** The tilt and yaw moments are visualized as a vector on the tilt-yaw plane. Output-constrained IPC then constrains this vector to a reference $r$. Shown here are the $\ell^\infty$ or $\ell^2$-norm of the load, which form the fundament of the $\ell^\infty$-IPC and $\ell^2$-IPC control methods.

the MBC transformation, namely $\ell^\infty$-IPC and $\ell^2$-IPC. It will become apparent that both methods require an estimation of the
original load, which is defined as the load that the system would experience without any IPC action, which is also derived.

### 4.1   Load norms

Using the MBC transformation, the oscillating load is decomposed in a tilt and a yaw contribution. These contributions are visualized using a vector in a tilt-yaw plane, shown in Figure 6. Conventional, full IPC drives this tilt-yaw moment vector to the origin, fully attenuating the 1P oscillating load. Output-constrained IPC instead drives this vector to a certain reference or
constrains the norm of this load vector to a certain value.

    In this work, the load is constrained using two different norms, the $\ell^\infty$ and $\ell^2$-norm, both defined through the p-norm

$$\left\| \begin{bmatrix} M_\mathrm{t} \\ M_\mathrm{y} \end{bmatrix} \right\|_p = \sqrt[p]{M_\mathrm{t}^p + M_\mathrm{y}^p}. \tag{11}$$

For $p = \infty$, this norm equals the infinity norm, resulting in a square reference, and for $p = 2$, this norm equals the Euclidian norm, resulting in a circular reference, both shown in Figure 6.

### 4.2   $\ell^\infty$-IPC

Using the $\ell^\infty$-norm of the load vector, the controller behaves in a decoupled manner and drives the tilt moment to a tilt reference $r_\mathrm{t}$ and the yaw moment to a yaw reference $r_\mathrm{y}$, as shown in Figure 7. It uses the decoupling between the tilt and yaw axes to use independent tilt and yaw controllers, defined as

$$C_\mathrm{t}(s) = C_\mathrm{y}(s) = \frac{K_\mathrm{i}}{s}, \tag{12}$$

where $K_\mathrm{i}$ is the integral gain.




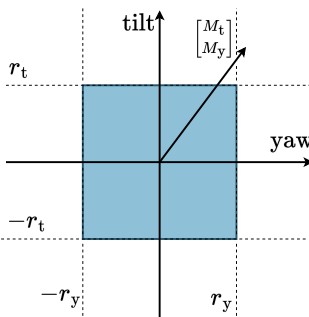

**Figure 7.** Using $\ell^\infty$-IPC, the nonrotating load vector is independently driven to a tilt and yaw reference $r_\mathrm{t}$ and $r_\mathrm{y}$. The load is thus constrained within an $\ell^\infty$-norm, indicated by a square.

This method was hypothesized in the introduction of Liu et al. (2022) as a "deadband" augmentation of conventional IPC and is an extension of Henry et al. (2024) by adding a yaw reference and the ability for the references to be both positive and negative.

The negative reference on the tilt axis is likely not often active. However, it may become necessary when the wind conditions
have a negative shear component, which does occur sometimes, especially offshore (Yan et al., 2022).

There are two main challenges with this approach. First, the tilt and yaw moments should never be amplified to the reference because the control objective is load reduction, not amplification. In closed-loop, when the controller drives the moment to a certain reference, the measurement of this moment alone is insufficient information to determine whether this load has been amplified or reduced to its current value. Second, the sign of the reference should be changed appropriately since the load
vector can take any value in the tilt-yaw plane, and can therefore switch signs. Similarly, once the closed-loop system is at a certain load, measuring the moments alone is insufficient information to determine whether the positive or negative reference should be used.

Both of these problems are solved by estimating the "original load", which is defined as the load in the tilt-yaw plane that the system would experience in open loop, so when the output-constrained IPC controller would be disengaged, the derivation
of the original load estimator will be discussed in Sect. 4.4

If the original tilt moment is positive, the tilt moment shall be constrained between its current value and 0. So a positive tilt reference is selected. Furthermore, the tilt controller is saturated on $[0, \infty)$, preventing the tilt moment from increasing so that the load is kept constant or reduced towards zero. If the reference is between the original tilt load and zero, the load is driven towards the reference, but if the reference is above the original tilt load, the controller is saturated and no IPC action is done.
If the original tilt moment is negative, the negative tilt reference is selected and the tilt controller is saturated on $(-\infty, 0]$, this allows the controller to increase the tilt moment, bringing any negative moment closer to the origin. The same logic applies to the yaw channel.

Figure 8 shows the block diagram for the $\ell^\infty$-IPC control method. The tilt and yaw moments, together with the tilt and yaw pitch angles are used to estimate the original load. The sign of the original tilt moment is then used to set the sign of the tilt



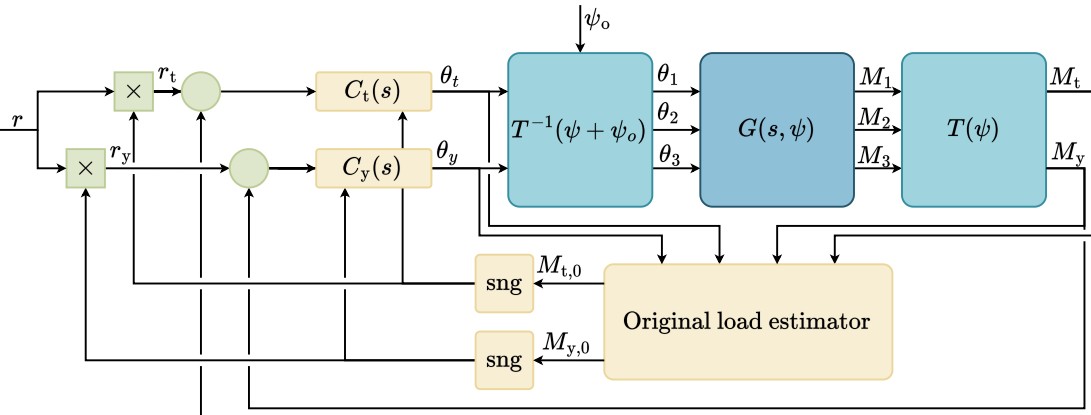

**Figure 8.** Block diagram of $\ell^\infty$-IPC. It uses a decoupled tilt and yaw controller that both drive the moment to the reference. To avoid load amplification, the reference sign and integrator saturation are adjusted based on the sign of the original load.

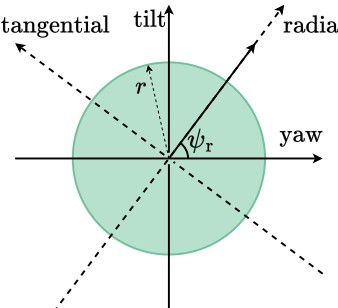

**Figure 9.** Using $\ell^2$-IPC, the nonrotating load vector is projected on a new orthogonal reference system, consisting of a radial and tangential axis. In this new axes system, the magnitude of the load is projected on the radial axes. The load is thus constrained using the 2-norm, indicated by a circle.

reference and select the saturation bounds for the tilt controller based on the logic mentioned previously. The same applies to the yaw channel. Overall, the tilt and yaw channels remain decoupled in this approach, and each channel either drives the load to its reference or is saturated to avoid amplification.

### 4.3 $\ell^2$-IPC

Instead of constraining the tilt and yaw loads separately, using the $\ell^2$-norm of the nonrotating load vector, the magnitude of the
load is constrained. This is achieved by projecting the nonrotating load vector to a new reference frame, where the magnitude of the load is projected on a single axis as shown in Figure 9. The new reference frame consists of a radial and a tangential component. In steady-state the magnitude of the load is projected on the radial axis and the tangential component would be



zero. This allows a single controller, defined as

$$C_r(s) = \frac{K_i}{s}, \tag{13}$$

to act on the radial axes to directly regulate the magnitude of the load. Note that the integral gain $K_i$ is equal to the integral gain of the $\ell^\infty$-IPC control method to provide a fair comparison. The rotation of this axes system is achieved using the rotations inherent to the MBC transformation, as previously discussed in Sect. 2.3. The rotation is done by angle $\psi_r$.

This means that the nonrotating pitch angles are in phase with the nonrotating load. This is ideal when the azimuth offset in the inverse transformation ensures that the phase lag from the nonrotating pitch to the nonrotating moment is zero. While the optimal azimuth offset is defined to achieve decoupling, it also causes an almost zero phase shift from the nonrotating pitch to the nonrotating moment.

Ensuring that the controller does not amplify the loads is more straightforward than for the $\ell^\infty$-IPC control method. Since the controller only acts on the magnitude, which is always positive, it should only lower the magnitude of the load, which is achieved by saturating the controller on $[0, \infty)$.

The challenge lies in estimating the rotation $\psi_r$ and its estimation bandwidth. It can be estimated by taking the inverse tangent of the tilt and yaw moment. However, when operating close to full IPC, these moments are reduced to zero, making this estimate noisy, or worse, undefined. As a solution, the $\ell^2$-IPC control method also uses the estimate of the original load for $\psi_r$, which is thus calculated as

$$\psi_r = \text{atan2}\left(M_{t,o}, M_{y,o}\right), \tag{14}$$

where $M_{t,o}$ and $M_{y,o}$ are the original tilt and yaw moments respectively, whose estimation will be discussed in Sect. 4.4. This ensures that a robust estimate of the rotation angle is obtained even when operating close to full IPC (and the measured load is close to zero).

The angle $\psi_r$ is effectively the phase of the final commanded pitch angles to the three blades. This phase should thus change sufficiently quickly to adapt to changes in wind conditions, but shall not contain noise as it directly translates to noise in the phase of the pitch signal.

Figure 10 shows the block diagram of $\ell^2$-IPC. Similarly to $\ell^\infty$-IPC, the original load estimator uses the nonrotating moments and pitch angles. The angle of the original load in the tilt-yaw plane is then calculated and used to rotate the tilt and yaw moment to a new reference system with a radial and a tangential component. The inverse rotation uses the same angle to rotate the radial control action back to a tilt and yaw component. This control method only needs a single controller, $C_r(s)$, to regulate the radial moment to a reference because it is actuating in the ideal phase. This is in contrast to conventional IPC and $\ell^\infty$-IPC, which control the tilt and yaw axis separately and thus need two controllers.

## 4.4 Original load estimation

The original load is the load experienced by the system if IPC were disengaged. In closed-loop, with IPC active, this cannot be measured and should instead be estimated. The estimation consists of a measurement of the current load and an estimation of the load that is regulated away with the current IPC control action, using a simple model.



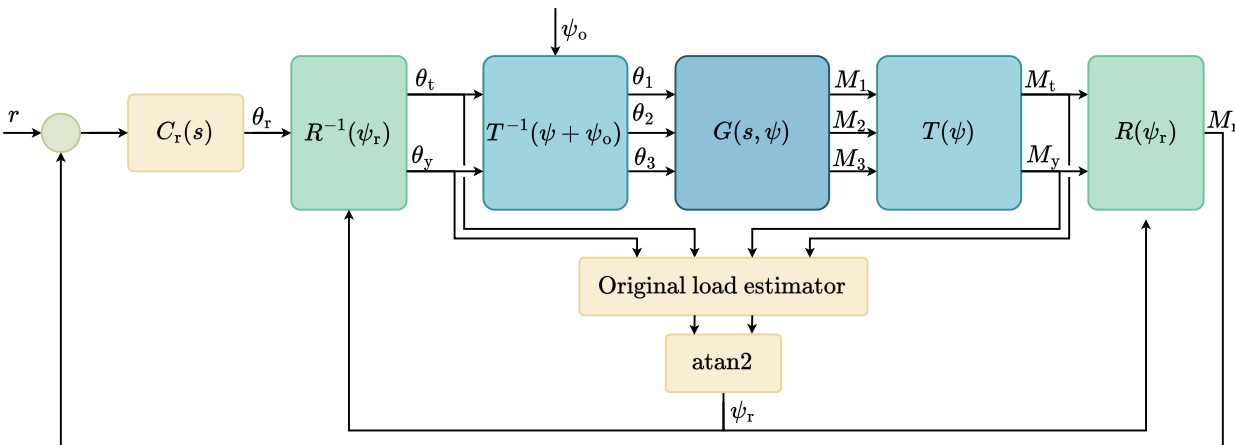

**Figure 10.** Block diagram of $\ell^2$-IPC. The original load estimator uses the nonrotating loads and nonrotating pitch angles to reconstruct the original load. The angle of this load in the tilt-yaw plane is $\psi_\mathrm{r}$, which is used to rotate the reference system to a radial and tangential axis where the magnitude of the nonrotating load vector is projected on the radial axis. A single SISO controller then regulates this load magnitude.

In steady-state, by using a first-order approximation, the original load is defined as

$$\boldsymbol{M}_{\mathrm{N,o}} = \boldsymbol{M}_\mathrm{N} - \mathbb{J}_{\boldsymbol{M}_{\mathrm{N,o}}}\boldsymbol{\theta}_\mathrm{N}, \tag{15}$$

with

$$\mathbb{J}_{\boldsymbol{M}_{\mathrm{N,o}}} = \begin{bmatrix} \frac{\partial M_0}{\partial \theta_0} & \frac{\partial M_0}{\partial \theta_\mathrm{t}} & \frac{\partial M_0}{\partial \theta_\mathrm{y}} \\ \frac{\partial M_\mathrm{t}}{\partial \theta_0} & \frac{\partial M_\mathrm{t}}{\partial \theta_\mathrm{t}} & \frac{\partial M_\mathrm{t}}{\partial \theta_\mathrm{y}} \\ \frac{\partial M_\mathrm{y}}{\partial \theta_0} & \frac{\partial M_\mathrm{y}}{\partial \theta_\mathrm{t}} & \frac{\partial M_\mathrm{y}}{\partial \theta_\mathrm{y}} \end{bmatrix},$$

which is the Jacobian of the nonrotating blade moments with respect to the nonrotating blade pitch angles and is equal to the steady state gain of $\tilde{G}(s)$. This estimate thus requires model information, which can be obtained through system identification or linearizations of a wind turbine model. Note that while this definition also includes the estimate of the original collective moment, only the original tilt and yaw moments are used by the $\ell^\infty$-IPC and $\ell^2$-IPC control methods.

Furthermore, due to the decoupling using the azimuth offset, the off-diagonal elements of $\mathbb{J}_{\boldsymbol{M}_{\mathrm{N,o}}}$ are much smaller than the

diagonal elements. So, similarly to the control, the estimation of the tilt and yaw original load is decoupled. This simplifies the estimation to

$$\begin{aligned} M_{\mathrm{t},o} &= M_\mathrm{t} - \frac{\partial M_\mathrm{t}}{\partial \theta_\mathrm{t}}\theta_\mathrm{t} \\ M_{\mathrm{y},o} &= M_\mathrm{y} - \frac{\partial M_\mathrm{y}}{\partial \theta_\mathrm{y}}\theta_\mathrm{y}. \end{aligned} \tag{16}$$

The two partial derivatives, $\partial M_\mathrm{t}/\partial \theta_\mathrm{t}$ and $\partial M_\mathrm{y}/\partial \theta_\mathrm{y}$ are almost equal to each other. This is also why the diagonal pitch controller can have the same gains for its tilt and yaw controller.





This estimate of the original load is a steady-state estimate and thus neglects the dynamics in the system. It is an instantaneous estimation of the original load after transient effects have died out. To reduce the bandwidth of this estimate and filter out high-frequency components, the estimate is filtered with a low-pass filter with a cutoff frequency of $\omega_\mathrm{o}$. This ensures that for the $\ell^\infty$-IPC controller, the switching between reference sign and integrator saturation is smooth and that for the $\ell^2$-IPC controller, no noise is introduced into the phase of the resulting sinusoidal pitch signal.

Furthermore, it is assumed that the Jacobian is constant around a certain operating condition and does not change as a function of the load reference. These two proposed output-constrained IPC controllers are tested and compared against the baseline in the following section.

## 5   Results

This section presents the results of the $\ell^\infty$-IPC and $\ell^2$-IPC control method and compares them to the baseline set in Sect. 3. It
also uses the same setup with the IEA 15 MW turbine, a wind speed of 15 m/s, and a vertical shear coefficient of 0.07 (except for case 3) in both laminar and turbulent conditions.

After controller calibration, four different cases are analyzed. First, in laminar wind conditions, the staircase response is shown to analyze the working mechanisms of the two control methods. Second, the trade-off between damage equivalent load of the flapping moment and actuator duty cycle is analyzed in laminar wind conditions. Third, the controllers are tested in
a wind field with changing shear coefficients to analyze the effectiveness of using the estimated original load for integrator saturation and reference sign selection for $\ell^\infty$-IPC and phase estimation for $\ell^2$-IPC. Lastly, both control methods were tested in turbulent wind conditions with multiple seeds to analyze the trade-off between damage equivalent load and actuator duty cycle in realistic wind conditions.

### 5.1   Controller calibration

The proposed output-constrained $\ell^\infty$-IPC and $\ell^2$-IPC control methods have the same variables that need to be calibrated, namely the azimuth offset, the integral gain for the IPC controller ($C_\mathrm{t}(s)$ and $C_\mathrm{y}(s)$ for the $\ell^\infty$-IPC controller and $C_\mathrm{r}(s)$ for the $\ell^2$-IPC controller), and the bandwidth and partial derivatives in the original load estimator. For a fair comparison, both control methods use the same calibration.

The optimal azimuth offset calculated in Sect. 3.2, found to be $24.5°$ is also used for $\ell^\infty$-IPC and $\ell^2$-IPC. In Sect. 3.3, a
grid search was performed for different crossover frequencies for conventional full IPC. For the $\ell^\infty$-IPC and $\ell^2$-IPC control methods, a crossover frequency of 0.2 rad/s is selected since the full IPC controller with this crossover frequency has a good trade-off between DEL and ADC, resulting in an integral gain $K_\mathrm{i} = -6.8 \cdot 10^{-7}$ for $C_\mathrm{t}(s)$, $C_\mathrm{y}(s)$, and $C_\mathrm{r}(s)$. Note that the controller gain is negative, while in conventional IPC controllers, this gain is positive. This is because the pitch controllers of $\ell^\infty$-IPC and $\ell^2$-IPC actuate based on the load errors rather than the load itself.

The bandwidth of the original load estimator is set to 2 rad/s. This allows for good noise suppression while simultaneously being fast enough to respond to changes in wind shear.





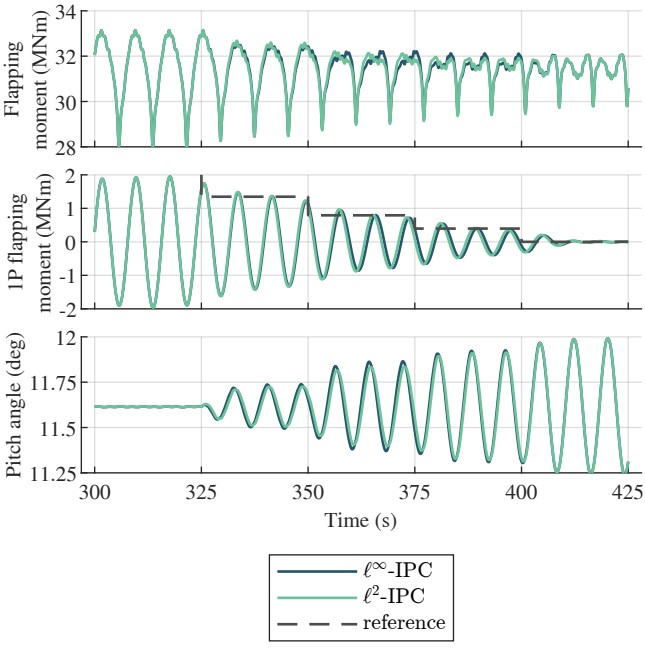

**Figure 11.** Time domain results to the staircase reference load input in the rotating frame. While both output-constrained IPC control methods follow the reference accurately, they have a slightly different control action and resulting flapping moment during the transition between no IPC to full IPC.

The partial derivatives in the original load estimator are estimated from the spectral estimate, also used to find the optimal azimuth offset. The tilt-to-tilt contribution, $\partial M_\mathrm{t}/\partial \theta_\mathrm{t}$, and the yaw-to-yaw contribution, $\partial M_\mathrm{y}/\partial \theta_\mathrm{y}$, are both equal to 109.3 dB.

### 5.2 Staircase response in laminar wind

The first case simulates the response of the controller in laminar conditions to analyze the working principles of the controllers and compare them. A staircase input on the reference is used to let the controller pass through different operating points between no IPC and full IPC. The reference load starts above the original load, resulting in no IPC control action from the output-constrained IPC controllers. It then steps to zero in four steps, each lasting 25 seconds, until the output-constrained IPC controllers perform equal to full IPC with a reference load of 0 Nm. The first 300 seconds are discarded to allow the system to
reach a steady state, and thus exclude initialization effects.

Note that the reference for the $\ell^\infty$-IPC control method is adjusted such that it achieves the same resultant load magnitude as the $\ell^2$-IPC control method.

Figure 11 shows the time domain results in the rotating frame of the flapping moment, 1P filtered flapping moment, and pitch signal. The 1P filtered signal is obtained by filtering the flapping moment with a finite impulse response (FIR) bandpass
filter with a passband between 0.75P and 1.25P. In the first subplot, the flapping moment initially shows a strong 1P component due to wind shear, and the effect of tower shadow is also clearly visible. As the reference is reduced, the 1P component

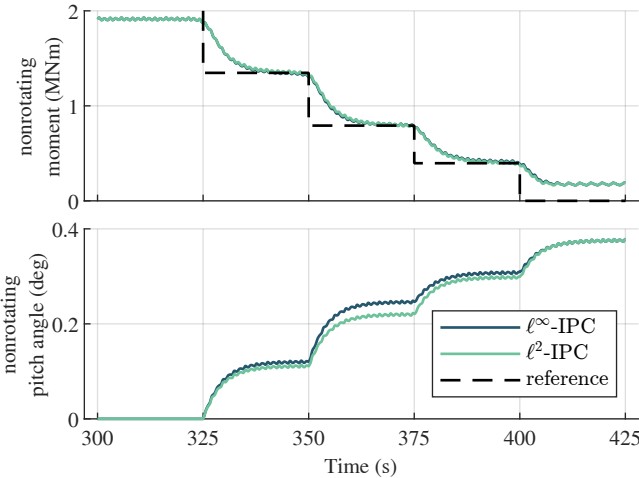

**Figure 12.** Time domain results to the staircase reference load in the nonrotating frame. Both control methods accurately follow the reference. The $\ell^2$-IPC controller requires a smaller nonrotating pitch magnitude, indicating that it is more efficient.

diminishes until the dominant oscillation is the 2P component. Furthermore, a small difference is observed between $\ell^\infty$-IPC and $\ell^2$-IPC, which is discussed more extensively later. In the second subplot, the peaks of the 1P flapping moment of the two output-constrained IPC control methods track the reference accurately. Crucially, the oscillation is not amplified towards the reference at the start of the simulation. Showing that the output-constrained IPC methods only activate when the load is above the reference. However, a small difference in phase between the two control methods can be observed. The third subplot also shows this phase shift also in the pitch signal. In addition, the $\ell^\infty$-IPC method requires a larger pitch magnitude to track the same reference as $\ell^2$-IPC. This occurs because the $\ell^\infty$-IPC method uses control effort to both change the magnitude and phase of the load, rather than only the magnitude.

The difference in control magnitude is more clearly visible in the nonrotating frame. Figure 12 shows the time domain results in the nonrotating frame of the nonrotating moment and nonrotating pitch angle. The three blade loads are transformed to the nonrotating frame using the MBC transformation and low-pass filtered. The controllers are compared by analyzing the magnitude of the tilt and yaw contribution of the nonrotating moment and pitch angle. Both controllers track the reference accurately and have a similar transient response due to their identical calibration. However, the $\ell^\infty$-IPC controller requires
a larger pitch magnitude to track the same reference as $\ell^2$-IPC. This is because the $\ell^\infty$-IPC controller uses control effort to both change the magnitude and phase of the load, rather than only the magnitude, as is clearly visible on the tilt-yaw plane, discussed next.

Figure 13 shows the tilt contribution on the y-axis and the yaw contribution on the x-axis. So the response over time becomes a line on the tilt-yaw plane. Both controllers start with a reference above the original load, resulting in no IPC action, thus equal
to no IPC. Once the reference is reduced in its first step, the $\ell^\infty$-IPC controller only drives the moment vector down in the tilt direction since the load is naturally already below the yaw reference. This happens again in the second step of the reference

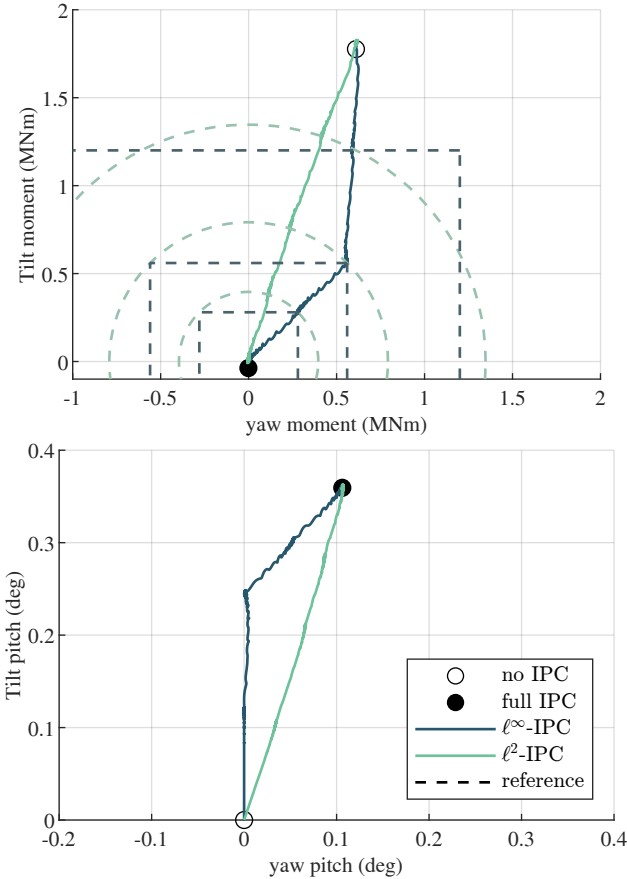

**Figure 13.** Time domain results plotted on the tilt-yaw plane for the staircase input. Both controllers start at no IPC and gradually go to full IPC in a few steps, resulting in a line in the tilt-yaw plane. The $\ell^\infty$-IPC control method initially only drives the tilt moment to the tilt reference and only starts actuating in yaw when the yaw moment is constrained by the yaw reference. On the other hand, the $\ell^2$-IPC method drives the load directly to the origin and keeps the same phase of the pitch action. By only spending control action on load reduction and not on a phase change of the load, $\ell^2$-IPC is more efficient.

load. Only when the tilt and yaw moment are equal and are both constrained by the reference, at a phase of $45°$, does the controller start actuating in yaw. The $\ell^2$-IPC controller, on the other hand, drives the load directly to the origin and keeps the same phase of the pitch action. The $\ell^2$-IPC control method is thus more efficient since it does not spend control effort to change the phase of the load in the tilt-yaw plane. So the $\ell^2$-IPC control method requires a smaller pitch magnitude to track the same reference as $\ell^\infty$-IPC. With a reference of zero, both controllers are equal in operation to full IPC.

Furthermore, a pure shear input does not lead to a pure tilt moment response. Even though the wind is strongest at the top of the tilt axis, the blade's dynamics cause the highest load to be experienced at the point indicated by no IPC.



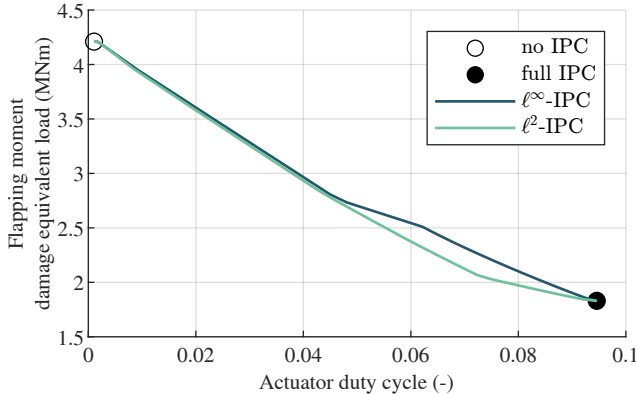

**Figure 14.** The trade-off when operating between no IPC and full IPC in laminar conditions. Especially close to full IPC, $\ell^2$-IPC achieves a larger reduction in the damage equivalent load of the flapping moment for the same actuator duty cycle than $\ell^\infty$-IPC.

Each reference thus leads to a separate steady-state point, which is used to analyze the trade-off between actuator effort and load reduction.

### 5.3   The trade-off in laminar conditions

The previous section showed that $\ell^2$-IPC achieves the same load reduction with a smaller pitch magnitude compared to $\ell^\infty$-IPC for a certain reference. This section expands on this by analyzing this trade-off for all operating points between no IPC and full IPC. For both control methods, 19 reference loads were analyzed, ranging from 0 Nm (full IPC) to 2000 Nm (above the original load, so no IPC). The load reduction is measured using the damage equivalent load (DEL, see Equation 9) while the actuator effort is measured using the actuator duty cycle (ADC, see Equation 10).

Again, the first 300 seconds from the simulation are discarded, and the last 100 seconds of each reference load are used to calculate the actuator duty cycle and damage equivalent load. Since the system operates in steady-state in laminar conditions, the DEL and ADC calculations do not require more data.

Figure 14 shows the trade-off between DEL and ADC for laminar conditions. For a reference load above the original load, both control methods are equal in operation to no IPC. When the reference load is reduced, initially both control methods roughly achieve the same reduction in DEL for an increase in ADC. However, as the reference load is further reduced, the $\ell^2$-IPC control method reduces the DEL up to 8.4% more than $\ell^\infty$-IPC. This is because the $\ell^2$-IPC controller actuates in the same phase as the load, while the $\ell^\infty$-IPC controller also changes the phase of the load, as previously shown in Figure 13, thus spending unnecessary control effort. As the reference load reaches zero, both controllers converge to full IPC. Furthermore, the slope of the trade-off between DEL and ADC diminishes close to full IPC, showing diminishing returns when opting for conventional full IPC.





Both proposed output-constrained IPC controllers can operate on any point between no and full IPC. In contrast, conventional IPC cannot trade off fatigue load and actuator effort in laminar conditions since the controller tuning does not affect the steady-state, as discussed in Sect. 3.3.

Note that this result depends on the shear coefficient and wind speed, and the difference between the two controllers can be larger or smaller depending on these parameters. For example, if the original load is a pure tilt moment, both controllers will have identical performance since the $\ell^2$-norm and $\ell^\infty$-norm are equal for a pure tilt moment. The following section evaluates the performance of the original load estimator when the wind shear coefficients are varying.

## 5.4 Varying the wind shear coefficients

The previous results were obtained for a constant vertical wind shear coefficient of 0.07. This section shows that both control methods work for time-varying horizontal and vertical shear coefficients, which could arise from a changing boundary layer or wake impingement from upstream wind turbines when operating in a wind park.

When the shear coefficients of the wind change, the $\ell^\infty$-IPC control method has to adjust the sign of the reference load and integrator saturation using the original load estimation, while the $\ell^2$-IPC has to adjust the phase of the pitch signal based on this estimate.

An artificial wind field was generated, starting with an initial vertical shear coefficient of 0.07 and a horizontal shear coefficient of 0. In two steps, the vertical shear coefficient reduces to 0 while the horizontal shear coefficient goes to -0.07. This is done in laminar wind conditions to analyze the working mechanisms of the original load estimator and its use by the two control methods. During the entire simulation, both control methods follow a constant reference of 1 MNm.

Figure 15 shows the results for the $\ell^\infty$-IPC control method. Only the tilt signals are shown, but the same analysis applies to the yaw signals. At 300 seconds, the $\ell^\infty$-IPC controller follows the reference of 1 MNm accurately using a positive pitch angle. Furthermore, the original tilt moment is accurately estimated since it is equal to the no IPC case, which does not do any individual pitch control and its tilt moment is thus equal to the original tilt moment. The first wind shear change occurs at 340 seconds, and the tilt moment naturally decreases below the reference. To amplify the tilt moment to the reference, the controller would require a negative tilt pitch action. However, since the original tilt moment is correctly estimated to be positive, the pitch action is saturated on $[0, \infty)$ so that the controller does not amplify the load towards 1 MNm. Since the tilt moment is naturally below the reference, the controller sets its tilt pitch signal to zero, furthermore highlighting the advantage of output-constrained IPC. After the second wind shear change, at 380 seconds, the original tilt moment is correctly estimated to be negative, so the pitch action is saturated on $(-\infty, 0]$. Simultaneously, the reference sign is flipped, so the controller is now driving the tilt moment to -1 MNm from an original tilt moment of about -1.4 MNm. This shows that in changing wind shear conditions, the $\ell^\infty$-IPC control method correctly sets its pitch saturation and reference sign by estimating the sign of the original load.

The $\ell^2$-IPC control method uses the estimate of the original load to estimate the phase $\psi_r$ of the original load, and then matches the phase of the pitch signal to this. By doing this, it achieves the ideal phase of the pitch signal. So the pitch signal phase of $\ell^2$-IPC should ideally match the pitch signal phase of full IPC. This is shown in Figure 16 for the wind field with

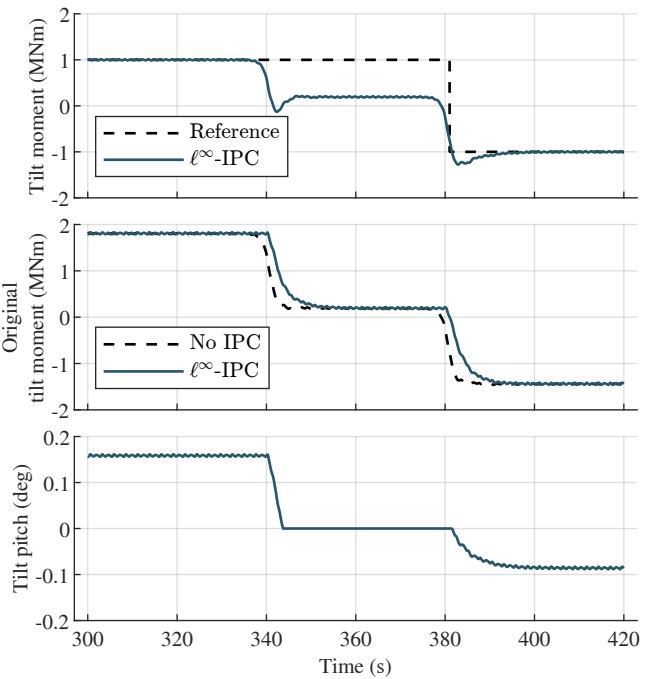

**Figure 15.** The $\ell^\infty$-IPC control method first follows a reference of 1 MNm accurately. At the first wind shear change, the load naturally goes below this reference and due to tilt pitch saturation, the controller does not amplify the load. Once its estimate of the original load becomes negative, just after the second wind shear change, the reference sign and the controller saturation change, allowing the controller to reduce the tilt moment to -1 MNm.

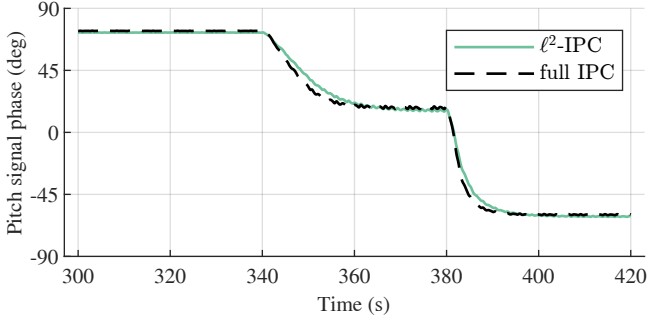

**Figure 16.** By estimating the phase of the original load, the $\ell^2$-IPC control method actuates in the ideal phase. The ideal phase is equal to the phase of the pitch signal of full IPC.

changing shear coefficients. In steady-state, the phase of the pitch signal accurately follows the ideal phase. In the transients, when the shear coefficients change, the phase is slightly different, which is due to the controller and estimator tuning. The next section analyses the controller performance in continually varying conditions using turbulent wind.



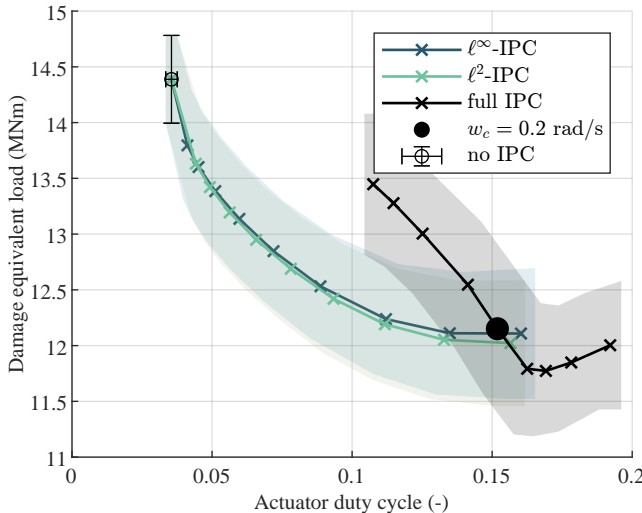

**Figure 17.** The trade-off when operating between no IPC and full IPC with 8.0 % turbulence. The $\ell^\infty$-IPC and $\ell^2$-IPC control methods use a change in reference load while the full IPC uses a grid search of integrator gains to achieve the trade-off in DEL and ADC. The output-constrained IPC methods have a nonlinear trade-off and achieve an 87% reduction in DEL at just 50% increase in ADC compared to the full IPC controller with the same crossover frequency (0.2 rad/s).

## 5.5 The trade-off in turbulent wind conditions

This section analyzes the effectiveness of the proposed output-constrained IPC methods in realistic turbulent wind conditions. To this end, the trade-off between DEL and ADC for laminar wind conditions, previously shown in Figure 14, is extended to turbulent wind conditions and compared to the baseline, discussed in Sect. 3.3.

Similarly to the baseline, the turbulent wind fields are generated using TurbSim (Jonkman, 2014) with 8% turbulence intensity for 10 different seeds, each with 30 minutes of useful data to obtain statistically significant results.

To analyze the different operating points between no IPC and full IPC, the $\ell^\infty$-IPC and $\ell^2$-IPC methods were run using multiple reference loads between 10 MNm (always above the original load, so no IPC) and 0 MNm (full IPC). Note that the no IPC load is substantially higher than for the laminar conditions due to load peaks due to turbulence.

Additionally, the data from the conventional, full IPC controller, as previously shown in Figure 5, is used to compare to the proposed control methods. Again, note that for full IPC a grid search for different crossover frequencies is used, while $\ell^\infty$-IPC and $\ell^2$-IPC only change their reference load and have a constant crossover frequency of 0.2 rad/s.

Figure 17 shows the trade-off between DEL and ADC in turbulent wind conditions and compares the $\ell^\infty$-IPC and $\ell^2$-IPC control methods to the baseline. Since each operating condition is run for 10 different turbulent wind seeds, there is a distribution of results. This distribution is assumed to be Gaussian, and the mean is shown as a line connecting the data points from each operating condition. The shaded region around this line represents one standard deviation from the mean. Since no IPC





has no parameters to tune, it only has a single mean and is thus shown as a data point with error bars representing the standard deviations.

First, both $\ell^\infty$-IPC and $\ell^2$-IPC smoothly transition between no IPC and full IPC by adjusting their reference loads. By setting the reference above the original load, the controllers both converge to no IPC while a reference of 0 MNm results in operation

close to the full IPC controller that uses the same crossover frequency, namely 0.2 rad/s. The trade-offs that $\ell^\infty$-IPC and $\ell^2$-IPC achieve are highly curved and do not go in a straight line from no IPC to full IPC. At the operating point where the ADC = 0.055, both methods achieve a 50% reduction in DEL with just 16.4% of the ADC, compared to full IPC tuned with the same crossover frequency. Furthermore, at ADC = 0.094, a 87% reduction in DEL with just 50% of the ADC. Thus, a significant decrease in actuator effort is achieved by aiming for a slightly lower reduction in fatigue loads using output-constrained IPC.

The relative change in fatigue load and actuator duty cycle are easier to observe in a normalized plot, given in Appendix A.

This is caused by the nonlinearity of the DEL calculation. Closer to no IPC, the controllers only attenuate the largest load peaks, which proportionally contribute more to the DEL than smaller peaks, which are only attenuated closer to full IPC, leading to diminishing returns.

Second, the difference between the two control methods is minimal in turbulent conditions, though $\ell^2$-IPC is slightly more

efficient when operating closer to full IPC. However, this advantage is well within one standard deviation. In laminar conditions, $\ell^2$-IPC was clearly more efficient because it did not change the phase of the load. The difference in phase between $\ell^\infty$-IPC and $\ell^2$-IPC is highest when the original load is $22.5° + k45°$ where $k \in \mathbb{N}$. But when the phase is $0° + k45°$, the $\ell^\infty$-IPC does not change the phase, and the two methods have an equal performance. In turbulent wind conditions, the phase of the original load is constantly changing, so the large advantage of $\ell^2$-IPC is not as pronounced anymore.

The $\ell^\infty$-IPC and $\ell^2$-IPC control methods were only run with a crossover frequency of 0.2 rad/s while the full IPC controller was run for a grid search over multiple crossover frequencies, as discussed in Sect. 3.3, and shown in black in Figure 17. When lowering the gain of the full IPC controller, it moves towards no IPC in a more straight line, not taking advantage of the nonlinearity of the DEL. Its lower gain ensures that it achieves a smaller load reduction, but the phase of the control action also starts to lag behind the optimal phase, thus reducing efficiency. The proposed output-constrained control methods do not suffer

from this effect since their load reduction and bandwidth are separated into the reference load set point and controller tuning. Furthermore, the full IPC controller can not reach the no IPC operating point by lowering the crossover frequency further because lowering its gain gets unpractical because the time constant of such a controller would be larger than 100 seconds, as discussed in Sect. 3.2.

The highest load reduction is still achieved by a full IPC controller with a crossover frequency of 0.75 rad/s, which is

higher than the crossover frequencies used for our proposed output-constrained controllers, namely 0.2 rad/s. By using the same crossover frequency for the $\ell^\infty$-IPC and $\ell^2$-IPC control methods and setting their reference to zero, they will achieve the same high level of load reduction. This shows that our methods are a natural extension to conventional IPC methods. The Pareto-optimal trade-off between DEL and ADC is likely a set of $\ell^2$-IPC controllers with different reference loads, crossover frequencies, and original load estimator bandwidths. This optimization is planned as future work.



## 6  Conclusions

In this work, we have explored the entire operating region between no and full IPC, using two newly proposed output-constrained IPC control methods, enabling the trade-off between actuator effort and load reduction, achieving an 87% reduction with just 50% of the actuation effort. The two methods, $\ell^\infty$-IPC and $\ell^2$-IPC, are natural extensions of the conventional IPC implementation based on the multiblade coordinate transformation, thus contributing to industry adoption.

In laminar conditions, both control methods accurately follow any load reference and thus operate on any point between full IPC and no IPC. In these conditions, $\ell^2$-IPC is more efficient since it does not use actuator effort to change the phase of the load. The original load estimator, used by the $\ell^\infty$-IPC control method to set its reference sign and integrator saturation and used by the $\ell^2$-IPC control method to set the phase of the control action, accurately reconstructs the original load using a steady-state estimate.

In turbulent wind conditions, both controllers can operate on any point between no IPC and conventional, full IPC. The trade-off is highly curved, and both control methods achieve a 50% load reduction, measured in damage equivalent load (DEL), with just 16.4% actuator effort, measured in actuator duty cycle (ADC), compared to full IPC with the same controller tuning. This is the most important result of this work, and it shows that not only do these controllers facilitate the trade-off between no and full IPC, but the new operating points also achieve an excellent trade-off between DEL and ADC. Since high ADC is a barrier for industrial application of IPC, these methods allow the industry to get most of the benefits with little downside.

In this work, both control methods are tuned with a crossover frequency of 0.2 rad/s, while in turbulent conditions, full IPC achieves the highest load reduction with a crossover frequency of 0.75 rad/s. Future work will optimally tune the proposed control methods to find the Pareto front between DEL and ADC by varying the reference load, crossover frequency, and original load estimation frequency.

Since these control methods make a smooth trade-off between no and full IPC, they will also be integrated into a wind turbine control co-design framework. The large reduction in fatigue loads with a small effect on actuator fatigue might lower the material costs of the blades at a small increase to the cost of the pitch actuation system, thus lowering the overall cost of wind turbines.

*Code and data availability.*  The code and data are available through Hummel (2024). Additionally, the code is available through GitHub[1] for easier access. However, the GitHub repository might be updated in the future.

## Appendix A:  The normalized trade-off in turbulent wind conditions

To normalize the results of Figure 17, no IPC is taken as 0% ADC increase and 0% DEL decrease, while the full IPC controller with a crossover frequency of 0.2 rad/s is taken as 100% ADC increase and 100% DEL decrease. Figure A1 shows this

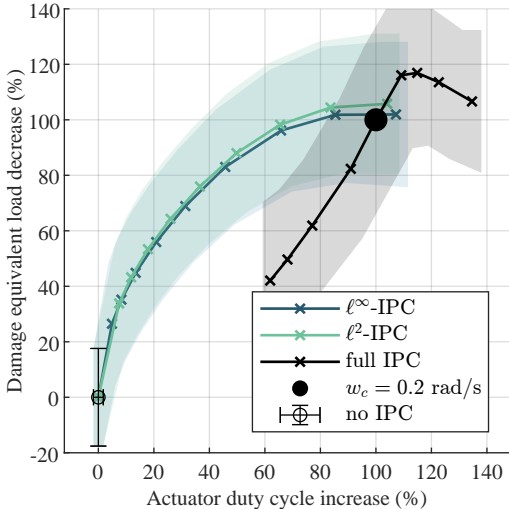

**Figure A1.** The normalized trade-off when operating between no IPC and full IPC with 8.0 % turbulence. The trade-off is normalized so that no IPC has 0% load reduction at 0% actuator effort increase while full IPC with a crossover frequency of 0.2 rad/s achieves 100% load reduction at 100% actuator effort increase. The two output-constrained methods achieve a relatively high amount of load reduction with a small actuator effort increase, compared to changing the crossover frequency of full IPC.

normalized trade-off. It is easier to see that both methods achieve an 87% reduction in DEL at just a 50% increase in actuator effort with respect to no IPC.

*Author contributions.* **J.I.S Hummel**: Analysis, conceptualization, methodology, software, visualization, writing - original draft preparation. **J. Kober**: Supervision, writing - review & editing. **S.P. Mulders**: Conceptualization, supervision, writing - review & editing.

*Competing interests.* The authors declare that they have no conflict of interest.

---

[1]github.com/jesseishi/Output-constrained-IPC



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
