# Peer review of "Output-constrained individual pitch control methods using the multiblade coordinate transformation: Trading off actuation effort and blade fatigue load reduction for wind turbines"

_Wind Energy Science, 2024_

## Referee Comment (RC1)

**1   General comments**

Overall, this is an interesting, well-structured, and promising paper that introduces two methods for constraining the load reduction capability of classical Coleman-transformation-based individual pitch control (IPC) to mitigate its impact on actuator usage. The core idea is to modify the reference signals of the conventional IPC controller—traditionally set to zero for both non-rotating axis components—by assigning an alternative reference value. This adjustment reduces the control error, thereby lowering the control effort required. The authors compare these methods to a more straightforward approach: scaling the controller gains in conventional IPC, which also reduces actuator effort at the cost of some load reduction capability. The study evaluates the proposed methods under both laminar and turbulent inflow conditions across various reference load constraints. Results indicate that these approaches allow for a reduction in actuator effort while preserving a significant portion of the load reduction benefits of classical IPC. Additionally, the proposed methods demonstrate an advantage over simple gain scaling by offering a more effective trade-off between actuator usage and load reduction.

**2   Individual scientific questions and comments:**

**2.1   General questions, discussions and comments:**

- In this study, the outputs of the controller (i.e., the pitch demands) are indirectly constrained by modifying their reference signal. However, this constraint remains somewhat loose, as actuator usage can still vary significantly depending on inflow conditions. In a future study, do you think the new reference values used in these methods could be defined as a fraction of the estimated loads or as an offset from the estimated (original) loads, rather than a fixed value? This approach would allow the controller's output to remain adaptive, adjusting its magnitude based on inflow conditions. At the same time, partial IPC, i.e. operating between full and no IPC, would consistently maintain the same proportional reduction in actuator effort across different inflow conditions.

- Could you comment more on how to select a load reference? Would it be a constant value for all the wind speeds (or wind speed bins)?

**2.2   Section specific comments:**

**i)   Section 1**

- The terms full IPC and no IPC are used before they are defined later in this section.

- It would be good if the authors commented more on the significance of reducing actuator duty cycle and its impact on pitch system wear.

**ii) Section 2.1**

- The statement "The periodicity of the blade load is caused by wind shear, tower shadow, and rotor misalignment" could be generalized, as these are not the only contributing factors. In addition to wind shear, tower shadow, and rotor misalignment, periodic blade loading can also result from wear, lateral misalignment (in the case of a waked turbine), static yaw, turbulence, or any other asymmetry in the inflow. I recommend broadening this expression to better reflect the range of possible influences.
- The statement "nP harmonics map to the nearest 3nP harmonics" is only valid for the 1P transformation. When using the 2P Coleman transformation, the loads are mapped to the harmonic at the rotating frame plus and minus 2P (e.g., 2P is mapped to 0P). I recommend updating the sentence accordingly.

**iii) Section 2.3**

- Azimuth offset is defined multiple times within the paper. In section 2.2, it is referred to as $\psi_{o,n}$ while, in section 2.3, it is referred to as $\psi_r$. I recommend a consistent notation for this parameter.

**iv) Section 2.5**

- Filtering $M_{N,n}$ would also introduce a phase delay (e.g., for low-pass filters). Does the azimuth offset calculation take this into account?

**v) Section 3.1**

- Max pitch rate seems a bit small. Any potential implications? Did the controller saturate for any of the tests?

**vi) Section 3.3**

- For Fig. 5, I recommend indicating the crossover frequencies, either directly on the plot or in the text. This would enhance the readability and clarity of the figure.
- In Fig. 5, what does the shaded (gray) area represent? Additionally, what do the bars in the no-IPC case indicate? I assumed they are similar to those in Fig. 17, but I recommend providing a description here as well for clarity.

**vii) Section 4.2**

- Based on the statement: "If the reference is between the original tilt load and zero, the load is driven towards the reference, but if the reference is above the original tilt load, the controller is saturated and no IPC action is taken." what are the implications of modeling inaccuracies in the load estimator? If the estimator

overestimates the loads, could the controller become saturated, preventing load reduction in certain instances, which is an undesirable outcome as discussed throughout the paper? This concern may be particularly relevant in co-design scenarios where components are sized under the assumption that the IPC controller remains active.

viii) Section 5

- On page 3, the phrase "… (except for case 3)" is used. What does case 3 refer to? I recommend introducing and explaining it before its first mention to improve readability.

ix) Section 5.3

- On page 22, you state "…showing diminishing returns when opting for conventional full IPC." Could you clarify what is meant by "returns" in this context? Additionally, for which method are these returns diminishing?

x) Section 5.4

- Could you comment also about the load reduction performance of $l^2$-IPC?

xi) Section 5.5

- For the marked points, I recommend indicating the load references and crossover frequencies, either directly on the plot or in the text, similar to Fig. 5. This would enhance the readability and clarity of the figure.

- When comparing different references for DEL and ADC, what do you mean by a 50% reduction in DEL? Could you clarify with respect to which quantity this 50% reduction is calculated? Similarly, for ADC, when stating "…16.4% of the ADC, …", could you specify what this percentage refers to?

- I assume that the reductions are reported relative to the performance of full IPC, indicating that ADC can be significantly reduced at the expense of sacrificing a small portion of the DEL reduction. However, if the no-IPC case is taken as the baseline, the relationship is reversed—achieving a significant reduction in DEL compared to no IPC requires maintaining most of the ADC effort used in full IPC.

**3  Technical corrections**

I recommend the following for improving readability:

- "leading to interactions between the tilt input and the yaw output, and vice versa" instead of "… so from the tilt input to the yaw output, and vice versa.", in page 8

- "Turbines with more flexible blades" instead of "Especially flexible wind turbines", in page 8
- "full IPC" (as it is already defined in introduction), "baseline IPC" or "conventional IPC" instead of "baseline full IPC", in page 10,
- "full IPC" (as it is already defined in introduction), "baseline IPC" or "conventional IPC" instead of "Conventional, full IPC", in pages 12, 13, 18, 22, 25, 27, since conventional (or baseline IPC) is always a full IPC.

---

## Author Comment (AC1)

Date April 25, 2025 Your reference wes-2024-153 Contact person Jesse Ishi Storm Hummel

> E-mail j.i.s.hummel@tudelft.nl Subject Author's Response

**Delft University of Technology**

Delft Center for Systems and Control

Reviewers Wind Energy Science Journal Address Mekelweg 2 (ME building) 2628 CD Delft The Netherlands

Dear reviewers,

We thank you for the constructive and thorough comments and suggestions for our paper. We believe that your feedback has helped us significantly improve the quality of the manuscript.

The objective of this document is to reply to the points raised and provide a detailed overview of the changes made. For each comment, a point-to-point response is provided in blue color, while the corresponding changes to the manuscript are reported in red. Please note that, in the enclosed marked-up version of the revised manuscript, the removed and added portions of the manuscript are indicated by red strikethrough text and blue underlined text, respectively. We believe this document provides clear and comprehensive responses to the reviewers' comments.

Yours sincerely,

Jesse Hummel Jens Kober Sebastiaan Mulders

Enclosure(s): General remarks Response to Reviewer 1 Response to Reviewer 2 Marked-up version of the revised manuscript

**General remarks**

After internal discussion and review, we have decided to implement the following changes, in addition to the changes based on the feedback from the reviewers.

- 1. Change 'original load' to 'open-loop load' throughout the paper as it more precisely and technically describes what we mean by it. But for consistency with the reviewer's comments, we have kept 'original load' in this document.
- 2. Added [1] as a citation that states that the 1P load is the dominant load that contributes to fatigue.

Date April 25, 2025 Page/of 3/14

**Response to reviewer 1**

**1 General comments** Overall, this is an interesting, well-structured, and promising paper that introduces two methods for constraining the load reduction capability of classical Coleman-transformation-based individual pitch control (IPC) to mitigate its impact on actuator usage. The core idea is to modify the reference signals of the conventional IPC controller—traditionally set to zero for both non-rotating axis components—by assigning an alternative reference value. This adjustment reduces the control error, thereby lowering the control effort required. The authors compare these methods to a more straightforward approach: scaling the controller gains in conventional IPC, which also reduces actuator effort at the cost of some load reduction capability. The study evaluates the proposed methods under both laminar and turbulent inflow conditions across various reference load constraints. Results indicate that these approaches allow for a reduction in actuator effort while preserving a significant portion of the load reduction benefits of classical IPC. Additionally, the proposed methods demonstrate an advantage over simple gain scaling by offering a more effective trade-off between actuator usage and load reduction.

**Response:** Thank you for your positive feedback and thorough review of our work. We appreciate your recognition of our contributions and the effort we put into preparing our manuscript.

**2 Individual scientific questions and comments:**

**2.1 General questions, discussions and comments:**

In this study, the outputs of the controller (i.e., the pitch demands) are indirectly constrained by modifying their reference signal. However, this constraint remains somewhat loose, as actuator usage can still vary significantly depending on inflow conditions. In a future study, do you think the new reference values used in these methods could be defined as a fraction of the estimated loads or as an offset from the estimated (original) loads, rather than a fixed value? This approach would allow the controller's output to remain adaptive, adjusting its magnitude based on inflow conditions. At the same time, partial IPC, i.e. operating between full and no IPC, would consistently maintain the same proportional reduction in actuator effort across different inflow conditions.

**Response:** Thank you for your questions and suggestions. Since it relates to the next point, we will discuss both points together below.

Could you comment more on how to select a load reference? Would it be a constant value for all the wind speeds (or wind speed bins)?
 Response: Thank you for your question and comment. Indeed, if a hard-constraint on the actuator usage is required, input-constrained IPC should be considered. Contrarily, if the maximum magnitude of the loads is the goal, our output-constrained controllers provide that. In a practical scenario, the two could maybe be combined. The advantage of output-constrained IPC that we see and highlight in the paper is that it will automatically phase itself in or out, depending on the wind conditions. Second, selecting a reference load can be subject to optimization over an expected set of wind conditions for the turbine's lifetime. Making the reference load a certain percentage of the estimated original load is a great suggestion. We think this is interesting future work and easy to implement due to the already existing original load estimator. We also see two challenges: it would effectively add an additional feedback loop, affecting the stability of the system and maybe more importantly lose the phase in/out behaviour of output-constrained IPC.

**Revised portion:** We have added a small elaboration in the introduction on when output-constrained IPC can be favoured over intput-constrained IPC. Additionally, we have added a direction for future work in the conclusion that explores the difference in performance to input-constrained IPC and explores a combined input and output-constrained IPC method as well as setting the reference load as a fraction of the estimated original load.

**2.2 Section specific comments:**

- i) Section 1
  - The terms full IPC and no IPC are used before they are defined later in this section.

**Response:** Thank you for your comment. We have added an explicit definition of no and full IPC and removed redundant naming as suggested by your third and fourth technical corrections.

Revised portion: Throughout the whole document.

It would be good if the authors commented more on the significance of reducing actuator duty cycle and its impact on pitch system wear.
 **Response:** Thank you for your comment, and we acknowledge that this link has been lacking in our initial submission. We have added a discussion with two additional citations on how IPC leads to excessive actuator wear [2] and how existing pitch systems also don't always have the thermal rating to continuously perform IPC [3].
 **Revised portion:** We have added this discussion to the part of the introduction that talks about field tests and the problem of higher pitch actuation.

- ii) Section 2.1
  - The statement "The periodicity of the blade load is caused by wind shear, tower shadow, and rotor misalignment" could be generalized, as these are not the only contributing factors. In addition to wind shear, tower shadow, and rotor misalignment, periodic blade loading can also result from wear, lateral

misalignment (in the case of a waked turbine), static yaw, turbulence, or any other asymmetry in the inflow. I recommend broadening this expression to better reflect the range of possible influences.

**Response:** We completely agree, so thank you for the suggestion for this improvement. We have adjusted the sentence to make it more general. **Revised portion:** The second paragraph in Section 2.1.

The statement "nP harmonics map to the nearest 3nP harmonics" is only valid for the 1P transformation. When using the 2P Coleman transformation, the loads are mapped to the harmonic at the rotating frame plus and minus 2P (e.g., 2P is mapped to 0P). I recommend updating the sentence accordingly.
 Response: Thank you for your comment. We see how we could have written this in a better way because we intended to talk about dynamics only in this section. Since the rotor rotates at 1P, only the 1P MBC transformation of the loads matter in this case. We have clarified this paragraph to make this clear and also changed the next paragraph to include that the MBC transformation can also be used at higher harmonics for control (as opposed to dynamics analysis).

Revised portion: Second and third paragraph in Section 2.1.

- iii) Section 2.3
  - Azimuth offset is defined multiple times within the paper. In section 2.2, it is referred to as  $\psi_{o,n}$  while, in section 2.3, it is referred to as  $\psi_r$ . I recommend a consistent notation for this parameter. **Response:** We agree with the reviewer about this inconsistency. Our intension was to make the notation in Section 2.3 consistent with  $\ell^2$ -IPC but we see now that sticking with  $\psi_{o,n}$  is better.

**Revised portion:** Changed  $\psi_r$  to  $\psi_o$  in section 2.3.

- iv) Section 2.5
  - Filtering  $M_{N,n}$  would also introduce a phase delay (e.g., for low-pass filters). Does the azimuth offset calculation take this into account? **Response:** Thank you for your comment. We believe that any filtering done in the nonrotating frame would not affect the coupling because the coupling is an effect from any phase delay in the rotating frame, arising from system dynamics. When looking at the demodulated plant in Figure 2, let's assume that we have succesfully decoupled the plant from  $\theta_{N,n}$  to  $M_{N,n}$ . If we now introduce additional filtering on  $M_{N,n}$ , there will be a larger phase delay between the input and output but not additional coupling because filtering  $M_{t,n}$  does not introduce coupling to  $M_{y,n}$ . We hope that this has sufficiently clarified your comment.
- v) Section 3.1
  - Max pitch rate seems a bit small. Any potential implications? Did the controller saturate for any of the tests?

Date April 25, 2025 Page/of 6/14

> **Response:** Thank you for your comment. We had not checked this beforehand and it turned out to be a very useful check. We found two types of violations. First, a few mild violations when the reference load was set at 500 or 0 kNm and second, some more severe violations. The second set only occured for the  $\ell^{\infty}$ -IPC control method when the reference was 500 or 0 kNm. In those cases, when the original load estimate crossed the 0 line, the saturation would change, effectively causing a step change in the pitch angle. Since we have not included any actuator dynamics, this caused a severe spike in the pitch rate. We have made two main changes to the text.

> **Revised portion:** Added a paragraph at the end of Section 4.2 where we discuss the robustness of this approach (also based on your later comment) and the potential for high pitch rates. Furthermore, we have added a paragraph at the end of Section 5 discussing the pitch rates experienced during all simulations.

- vi) Section 3.3
  - For Fig. 5, I recommend indicating the crossover frequencies, either directly on the plot or in the text. This would enhance the readability and clarity of the figure.

**Response:** We have combined the response to this comment with the response to the comment below.

In Fig. 5, what does the shaded (gray) area represent? Additionally, what do the bars in the no-IPC case indicate? I assumed they are similar to those in Fig. 17, but I recommend providing a description here as well for clarity.
 **Response:** Thank you for your comments regarding Fig. 5. To address them, we have updated both the figure and the caption. To avoid cluttering the figure, we have only updated the legend to include a description of the shaded area, which is the standard deviation (1-sigma) over the 10 different simulations. We have updated the caption to explain that for both no IPC and full IPC we are showing a distribution, where the errorbar/shaded area represents the 1-sigma deviation. We have also added clarification to the crossover frequencies used.

**Revised portion:** The caption of Figure 5.

- vii) Section 4.2
  - Based on the statement: "If the reference is between the original tilt load and zero, the load is driven towards the reference, but if the reference is above the original tilt load, the controller is saturated and no IPC action is taken." what are the implications of modeling inaccuracies in the load estimator? If the estimator overestimates the loads, could the controller become saturated, preventing load reduction in certain instances, which is an undesirable outcome as discussed throughout the paper? This concern may be particularly relevant in co-design scenarios where components are sized under the assumption that

Date April 25, 2025 Page/of 7/14

the IPC controller remains active.

**Response:** Thank you for your comment. We believe that the control scheme is fairly robust against modeling errors and agree that this is a nice point to touch upon explicitly in the paper. The  $\ell^{\infty}$ -IPC controller changes its behaviour (the sign of the reference and its saturation), based on the sign of the original load. So problems can arise when the actual original load is negative but the estimated original load is positive. We believe this can happen if the original load is fairly close to zero and the reference is also zero. **Revised portion:** Added a paragraph at the end of Section 4.2.

- viii) Section 5
  - On page 3, the phrase "... (except for case 3)" is used. What does case 3 refer to? I recommend introducing and explaining it before its first mention to improve readability.
     Response: Thank you for your comment. We have removed the reference to case 3 which is only introduced later. And used different wording to indicate

case 3 which is only introduced later. And used different wording to indicate that one of the cases uses a different wind shear coefficient. **Revised portion:** First paragraph of Section 5.

- ix) Section 5.3
  - On page 22, you state "... showing diminishing returns when opting for conventional full IPC." Could you clarify what is meant by "returns" in this context? Additionally, for which method are these returns diminishing?
     **Response:** Thank you for your comment. After rereading these sentences, we agree that they could be better formulated. We have rewritten and extended the last sentence to indicate that as both controllers operate closer to full IPC, their reduction in DEL for a given increase in ADC becomes smaller.
     **Revised portion:** Changed a paragraph in Section 5.3.
- x) Section 5.4
  - Could you comment also about the load reduction performance of l2-IPC?
     Response: Thank you for your suggestion. We have changed Figure 16 to be similar to Figure 15 and the accompanying text. After making the edit, we appreciate how this helps show the difference between the two controllers.
     Revised portion: Figure 16 and the text discussing Figure 16.
- xi) Section 5.5
  - For the marked points, I recommend indicating the load references and crossover frequencies, either directly on the plot or in the text, similar to Fig. 5. This would enhance the readability and clarity of the figure.
     **Response:** Thank you for your comment. We have updated Figure 17 and Figure A1 to be consistent with our changes to Figure 5. To avoid clutter, we have opted to include the load references and crossover frequencies in the text, rather than in the figures themselves. Furthermore, we have simplified

the plots by showing only full IPC with a crossover frequency of 0.2 rad/s to compare against.

**Revised portion:** Figure 17, its caption, and one line in the discussion of this Figure.

 When comparing different references for DEL and ADC, what do you mean by a 50% reduction in DEL? Could you clarify with respect to which quantity this 50% reduction is calculated? Similarly, for ADC, when stating "...16.4% of the ADC, ...", could you specify what this percentage refers to?
 Response: Thank you for your comment. All relative comparisons are made with respect to the difference between no and full IPC. We have updated the wording in the caption and text to clarify this.
 Revised portion: Caption of Figure 17 and surrounding text

Revised portion: Caption of Figure 17 and surrounding text.

• I assume that the reductions are reported relative to the performance of full IPC, indicating that ADC can be significantly reduced at the expense of sacrificing a small portion of the DEL reduction. However, if the no-IPC case is taken as the baseline, the relationship is reversed—achieving a significant reduction in DEL compared to no IPC requires maintaining most of the ADC effort used in full IPC.

**Response:** Thank you for your comment. Indeed, our work focuses on how output-constrained IPC can operate between no and full IPC. To that end, we report the relative changes between no and full IPC. So we look at the relative changes with no IPC as the baseline. To clarify this, we have included an equation in the appendix to show how these percentages are calculated. **Revised portion:** Added text and an equation in Appendix A.

**3** Technical corrections**

I recommend the following for improving readability:

- "leading to interactions between the tilt input and the yaw output, and vice versa" instead of "... so from the tilt input to the yaw output, and vice versa.", in page 8
- "Turbines with more flexible blades" instead of "Especially flexible wind turbines", in page 8
- "full IPC" (as it is already defined in introduction), "baseline IPC" or "conventional IPC" instead of "baseline full IPC", in page 10,
- "full IPC" (as it is already defined in introduction), "baseline IPC" or "conventional IPC" instead of "Conventional, full IPC", in pages 12, 13, 18, 22, 25, 27, since conventional (or baseline IPC) is always a full IPC.

**Response:** Thank you for these technical corrections. We have adopted them throughout the document.

Revised portion: Whole document.

**Response to reviewer 2**

This paper proposes two MBC-based IPC techniques to achieve a trade-off between DEL reduction and pitch actuation by using a reference tracking IPC based on the estimation of original (non-IPC) blade loads. The I-infinity method uses individual controllers to mitigate tilt and yaw axes loads separately, whereas the I-squared technique projects the magnitude of the blade loads on to the radial axis and uses a single controller to reduce blade loads. The two controllers are compared against a baseline at 15 m/s for laminar and turbulent flows, varying reference loads and varying horizontal and vertical shear.

Overall, this is a thorough and well-written paper that makes a worthy contribution to wind turbine control research. The paper is structured well and easy to follow along. Below are some suggestions that may further improve the quality of the paper.

**Response:** Thank you for your review. We are happy to read your recognition of our work and are pleased to further improve our manuscript using the provided points.

1. The paper provides a sufficient literature review of input-constrained, outputconstrained, and fatigue-constrained IPC techniques. It also clearly differentiates the proposed work from the literature and quantifies the performance of the proposed controllers. However, it is difficult to place the performance of the proposed control techniques with respect to those in the literature review. Providing the key metrics from past research that are comparable to the proposed techniques will help place performance in perspective.

**Response:** Thank you for your comment. We appreciate your acknowledgment of our literature review and positioning of our work. While we agree that a performance comparison from past research would provide additional context, the data to do this is unavailable. As mentioned in the introduction, we are unaware of any constrained IPC papers that explore the complete trade-off between no and full IPC, and made this a key contribution of our work. Furthermore, the primary aim of this paper is to introduce two novel control methods rather than to optimize their performance. We have added a recommendation to do a simulation study to compare the performance of different contrained IPC methods.

**Revised portion:** Recommendation for future work in the conclusion.

2. It would help to comment on the robustness of the proposed techniques. Especially, the robustness of the original load estimator to varying turbulence intensity, varying wind speed and varying wind shear. In particular, is the Jacobian in equation 15 dependent on wind speed or collective blade pitch angle? What would be the required procedure if the controller is to be designed for the entire full-load operating region as opposed to a single wind speed. Does higher wind turbulence than what was tested affect the performance of the original load estimator and the

reference sign/angle output? While horizontal and vertical wind shear is varied, it is not clear if low overall wind shear is tested. In particular, is equation 14 stable when the original tilt and yaw moments are non-zero but small.

**Response:** Thanks for your comments, they show a deep understanding of our method and helped us to strengthen our contribution. The original load estimator basically uses the steady-state effect of the blade pitch on the blade load and should thus be scheduled similarly to a collective pitch controller, so with wind speed (or blade pitch). We don't think that there is a dependence on turbulence level, since this does not influence the steady-state behaviour of the blade. To further clarify this aspect, we have added a discussion in the original load estimator section. When the original tilt and yaw moments are non-zero but small and oscillating around the origin, more noise will be introduced to the calculation of the load rotation  $\psi_r$  (equation 14). However, no control effort would have to be spent to reduce the loads, since they are already small. If the loads are small and oscillating around a certain point, there will be some noise in the load rotation  $\psi_{
m r}$ . If the load reference is still set at zero, this noise will propagate in the control signal, but we don't think instability will occur. We did however observe some step changes in commanded pitch when the original load would make a zero-crossing and have added a discussion on this at the end of the results section.

**Revised portion:** Added a paragraph to Section 4.2 and 4.4 to discuss the robustness of the open-loop estimator and its need to be gain-scheduled. Furthermore, we have added a paragraph at the end of the results section that talks about those observed step changes.

Minor comments:

1. Line 78: Unclear what the difference between pitch actuation and actuator activity is.

**Response:** Thank you for catching this error. It should've been blade fatigue and actuator activity. Furthermore, we noticed that Collet et al. have published a journal paper [4] based on this work so we have updated our citation to their journal paper.

**Revised portion:** Changed pitch actuation to blade fatigue in the introduction.

- Lines 126-129: the mapping of nP harmonics is confusing, please elaborate on this. Response: Thank you for your comment. We have rewritten this paragraph based on your feedback and the feedback of reviewer 1. Revised portion: Section 2.1.
- 3. Line 138: The M\_b being bold is confusing as M\_b is a scalar component of M\_R (line 144)

**Response:** Thank you for your comment. We meant to stick to that convention too so thanks for catching this oversight. We have adjusted it accordingly. **Revised portion:** Changed  $M_{\rm b}$  to normal font in Section 2.2.

4. Line 155: denotes pitch angles in non-rotating frame sound repetitive within the sentence.

**Response:** Thank you for your comment and catching this repetition. **Revised portion:** We have removed one of the repetitive references to the non-rotating frame in Section 2.2.

- Line 164: Please confirm the T-1 transform is correct with all cosines.
   Response: Thank you for catching this error.
   Revised portion: In equation 3 the third columnn now contains sines.
- 6. Line 216: subscript is omitted can be mentioned earlier? **Response:** Thank you for your comment. Before starting with the writing section 2, we thought about either focusing on only 1P control, and thus omitting the \_n subscript, or keeping the section a more general introduction into IPC using MBC. We ended up choosing the latter option.
- In Fig 5: Clarify what the bands in the legend or the figure description, this is clarified to be one std dev. much later in the text.
   Response: Thank you for your comment. We have added this to the caption.
   Revised portion: Figure 5 caption.
- Line 283: For this to be true the units in Fig. 5 should be MNm?
   Response: Thank you for catching this error. Indeed, the units in Fig. 5 should be MNm.

Revised portion: Figure 5 y-axis label unit changed to MNm

9. Line 286: Comment on why the DEL increases with higher bandwidth? **Response:** Thank you for your suggestion. We investigated this, and due to the smaller stability margins on higher bandwidth controllers, the response was more oscillatory, resulting in a higher DEL. We have added this explanation to the manuscript.

**Revised portion:** Added an explanation to the last paragraph of section 3 and the caption of Fig. 5.

10. Line 345: radial axis\* ?

**Response:** Thank you for catching this typo. We have corrected it here, as well as in the caption of Fig. 9.

**Revised portion:** Changed 'axes' to 'axis' in the text and in the caption of Figure 9.

- Line 462: There are two single-line paragraphs here.
   **Response:** Thank you for the suggestion. We agree that combining these two paragraphs improves the appearance of the text.
   **Revised portion:** Combined the last two paragraphs in Section 5.2.
- Line 481: The diminishing slope is hard to see, maybe mention or provide a zoomed overlay in Fig. 14
   Perspense: Thank you for your comment. I think we have wrongly used the word

Response: Thank you for your comment. I think we here wrongly used the word

Date April 25, 2025 Page/of 13/14

diminishing and have now replaced it with 'reduces in steepness'. **Revised portion:** Rewrote a paragraph in Section 5.3 to replace the wording diminishing returns.

Date April 25, 2025 Page/of 14/14

[revised manuscript text omitted]

(2)

165

where  $\theta_{N,n} = \begin{bmatrix} \theta_{0,n} & \theta_{t,n} & \theta_{y,n} \end{bmatrix}^T$  denotes the nonrotating pitch vector consisting of the collective, tilt, and yaw pitch angles of the *n*P harmonic in the nonrotating frameand denotes the pitch angles in the nonrotating frame. The IPC controller is only active on the tilt and yaw channels, which relate to the oscillating part of the moments, so the first row and column are filled with zeros. These controllers fully attenuate the tilt and yaw moments by producing the necessary tilt and yaw pitch angles to drive the tilt and yaw moments to zero. In this work, we refer to this as unconstrained, conventional, or full IPC.

These tilt and yaw pitch angles are then converted back to the rotating frame using the inverse MBC transformation, defined 170 as

$$\boldsymbol{\theta}_{\mathrm{R}}(t) = \mathbf{T}_{n}^{-1}(\psi(t) + \psi_{\mathrm{o},n})\boldsymbol{\theta}_{\mathrm{N},n}(t), \tag{3}$$

with

$$\mathbf{T}_{n}^{-1} = \begin{bmatrix} 1 & \cos(n[\psi_{1}(t) + \psi_{\mathrm{o},n}]) & \sin(n[\psi_{1}(t) + \psi_{\mathrm{o},n}]) \\ 1 & \cos(n[\psi_{2}(t) + \psi_{\mathrm{o},n}]) & \sin(n[\psi_{2}(t) + \psi_{\mathrm{o},n}]) \\ 1 & \cos(n[\psi_{3}(t) + \psi_{\mathrm{o},n}]) & \sin(n[\psi_{3}(t) + \psi_{\mathrm{o},n}]) \end{bmatrix},$$

where  $\boldsymbol{\theta}_{\mathrm{R}} = \begin{bmatrix} \theta_1 & \theta_2 & \theta_3 \end{bmatrix}^{\mathsf{T}}$  denotes the rotating pitch vector containing the pitch angles of the three blades defined in the rotating frame and  $\psi_{\mathrm{o},n}\in\mathbb{R}$  denotes the azimuth offset used for the  $n\mathrm{P}$  harmonic. 175

**2.3 **Rotations in the MBC transformation**

The MBC transformation can be decomposed as a Clarke transformation followed by a rotation (O'Rourke et al., 2019). Note that the authors define this rotation as the DO0 transformation, but the DO0 transformation is usually defined as equal to the MBC transformation.

180

An offset in the forward or inverse MBC transformation results in an offset in the rotation transformation and thus a rotation of the nonrotating frame. This is mathematically derived by considering the forward MBC transformation for the first harmonic with an azimuth offset  $\psi_{\rm T} \psi_{\rm Q}$ , given by

$$\mathbf{T}(\psi + \psi_{\underline{r}\,\mathbf{o}})$$

$$= \frac{2}{3} \begin{bmatrix} 1/2 & 1/2 & 1/2 \\ \cos(\psi_1 + \psi_0) & \cos(\psi_2 + \psi_0) & \cos(\psi_3 + \psi_0) \\ \sin(\psi_1 + \psi_0) & \sin(\psi_2 + \psi_0) & \sin(\psi_3 + \psi_0) \end{bmatrix}$$
$$= \frac{2}{3} \begin{bmatrix} 1 & 0 & 0 \\ 0 & \cos(\psi_0) & -\sin(\psi_0) \\ 0 & \sin(\psi_0) & \cos(\psi_0) \end{bmatrix}$$
$$\cdot \begin{bmatrix} 1/2 & 1/2 & 1/2 \\ \cos(\psi_1) & \cos(\psi_2) & \cos(\psi_3) \\ \sin(\psi_1) & \sin(\psi_2) & \sin(\psi_3) \end{bmatrix}$$
$$= R(\psi_{\underline{r}o}) \mathbf{T}(\psi),$$

where  $\frac{R(\psi_r)}{P_r} R(\psi_0)$  is a rotation matrix rotating the nonrotating frame by  $\psi_r \psi_0$  around the collective axis. A similar derivation 185 holds for the inverse MBC transformation.

**2.4 Frequency domain analysis**

The MBC transformations can be transformed to the Laplace domain, enabling frequency domain analysis, useful for controller design and calibration. The main results from Lu et al. (2015) and Mulders et al. (2019) are given here. Transforming the

(4)

Figure 2. Block diagram of the demodulated plant  $\tilde{G}_n(s)$ . By calibrating the azimuth offset  $\psi_{o,n}$ , the low-frequency cross-coupling from  $\theta_t$  and  $\theta_y$  to  $M_y$  and  $M_t$  respectively, is minimized.

forward transformation (Eq. (1)) to the Laplace domain yields

190
$$\mathcal{M}_{N,n}(s) = \frac{2}{3}C_{L,n}\mathcal{M}_{R}(s_{-}) + \frac{2}{3}C_{H,n}\mathcal{M}_{R}(s_{+}),$$
 (5)

and the inverse transformation (Eq. (3)) is transformed to

$$\boldsymbol{\theta}_{\mathrm{R}}(s) = \tilde{C}_{\mathrm{L},n}^{\mathsf{T}}(\psi_{\mathrm{o},n})\boldsymbol{\theta}_{\mathrm{N},n}(s_{-}) + \tilde{C}_{\mathrm{H},n}^{\mathsf{T}}(\psi_{\mathrm{o},n})\boldsymbol{\theta}_{\mathrm{N},n}(s_{+}),\tag{6}$$

where  $C_{L,n}$  and  $C_{H,n}$  denote the *low* and *high* partial transformation matrices for the *n*P harmonic respectively,  $\tilde{C}_{L,n}^{\mathsf{T}}(\psi_{o,n})$  and  $\tilde{C}_{H,n}^{\mathsf{T}}(\psi_{o,n})$  the transpose of the partial transformation matrices including azimuth offset for the *n*P harmonic, and  $s_{\pm} = s \pm jn\omega_{r}$  and denotes the frequency-shifted Laplace operator.

By analyzing the wind turbine plant  $G(s, \psi)$  surrounded by the MBC transformations, the demodulated plant  $G_n(s)$  for the *n*P harmonic is defined, shown in Figure 2. This demodulated plant is linear time-invariant (LTI) after averaging out the azimuth dependency, which physically makes sense since the rotating blade loads are transformed to a nonrotating frame, where the azimuth angle has no meaning (Bir, 2008). Note that the demodulated plant is depicted as a 2 × 2 system without the collective input  $\theta_0$  and collective output  $M_0$  since these are not linked to the periodic loads and are not used by the individual

200 collective input pitch controller.

195

Due to dynamics in the wind turbine, there is a coupling between the tilt and yaw inputs and outputs. The system thus needs to be decoupled to enable the use of SISO controllers.

**2.5 Decoupling using the Optimal Azimuth Offset**

205 Phase lag in the system, due to, e.g., actuator dynamics, blade dynamics, dynamic induction, and communication delays, causes coupling in the nonrotating frame, so from leading to interactions between the tilt input to and the yaw output, and vice versa. Especially flexible wind turbines Turbines with more flexible blades have slower dynamics, and thus more coupling, which must be considered during the controller design process.

The literature describes two ways to deal with this coupling. The first is to design a multivariable controller that takes the coupling into account (Bossanyi, 2003; Lu et al., 2015). Second, the system can be decoupled by using an azimuth offset in the inverse MBC transformation (Mulders et al., 2019; Mulders and van Wingerden, 2019). After the system has been decoupled, SISO controllers are used to independently control the tilt and yaw moments.

The demodulated plant  $\tilde{G}_n(s)$  is fully decoupled when its relative gain array (RGA) (Skogestad and Postlethwaite, 2001) is close to identity. The RGA is a measure of the coupling between the inputs and outputs of a system and is defined as

215
$$\mathbf{R}(j\omega) = \mathbf{H}(j\omega) \circ \mathbf{H}(j\omega)^{-\mathsf{T}},$$
 (7)

where H denotes the frequency response matrix of the system,  $\omega$  the frequency, and  $\circ$  element-wise multiplication.

The azimuth offset allows the system to be decoupled in the low-frequency region and works by introducing an offset between the forward and inverse rotations, discussed in Sect. 2.3. Decoupling at low frequencies is sufficient and effective because the IPC controller is only active in this region. The optimal azimuth offset is found by minimizing the RGA of the off-diagonal components for low frequencies. In this work, the level of low-frequency coupling is defined as the highest off-diagonal element of the RGA matrix averaged over the low frequencies and is given by

$$R_{\#} = \max_{m \neq n} \left\{ \frac{1}{\omega_{\mathrm{m}}} \int_{0}^{\omega_{\mathrm{m}}} |\mathbf{R}_{m,n}(\mathbf{j}\omega)| \,\mathrm{d}\omega \right\}, \qquad m, n \in \{\mathbf{t}, \mathbf{y}\},$$
(8)

[revised manuscript text omitted]